# MAX-MIN SLICED GROMOV-WASSERSTEIN

## ABSTRACT

The Gromov-Wasserstein (GW) distance is a powerful tool for comparing objects across different metric spaces, but its high computational complexity limits its applicability. Although the Sliced Gromov-Wasserstein (SGW) discrepancy addresses this issue by projecting onto 1D distributions, it sacrifices key isometric properties such as reflection and rotation invariance. In this work, we introduce the max-min Sliced Gromov-Wasserstein (MSGW), a new variant that preserves the computational efficiency of SGW while ensuring essential isometric properties. This metric can be viewed as an adversarial game and is closely tied to the Hausdorff distance. Empirical results demonstrate that MSGW achieves competitive performance with a limited number of projections and excels in scenarios with varying dimensions, making it a practical and robust alternative to existing methods. Our anonymous implementation is available at https://anonymous.4open.science/r/MSGW-iclr2026-7600.

## 1 INTRODUCTION

In various applications within machine learning and data science, objects such as point clouds, images, graphs, and documents can be represented as probability measures by normalizing their relevant weights or frequencies. This modeling approach enables the comparison of these objects through metrics and methods designed to measure differences between probability measures. Optimal Transport (OT), or, based on it, the Wasserstein distance, has been widely used as a method to compare such objects in numerous domains. However, while the Wasserstein distance is effective within the same metric space, it does not allow us to define a loss function across different spaces. It thus fails to compare objects that are best described in different metric spaces. To address this limitation, the Gromov-Wasserstein (GW) distance was introduced (Mémoli, 2011; Peyré & Cuturi, 2019). The GW distance relies on the intra-relationships within each object's space, enabling cross-space comparisons. Both Wasserstein and GW distance have proven to be powerful in a wide range of tasks, including generative models (Arjovsky et al., 2017; Salimans et al., 2018), image processing Papadakis (2015), graph alignment (Vayer et al., 2019a; Pan et al., 2024), with real-world use cases in genomics (Demetci et al., 2020), and quantum chemistry (Peyré et al., 2016).

Despite the advantages of the GW distance, its computational difficulty is a long-standing problem. While the original Wasserstein distance involves a linear program, the GW distance defines a non-convex quadratic assignment problem, which is NP-hard in general. In this work, we introduce a novel computational method for solving the GW problem, based on the idea of sliced OT. Sliced OT has been particularly useful for high-dimensional datasets by projecting them onto one- or low-dimensional subspaces (Zhang et al., 2023). The approach has been applied accordingly for the Wasserstein distance (Julien et al., 2011; Bonneel et al., 2015; Paty & Cuturi, 2019), and also to the GW distance (Vayer et al., 2019b). However, previous sliced GW methods are either computationally prohibitively expensive or lose some of the structural properties of the original GW distance. We address both these issues by incorporating a max-min optimization in the sliced GW problem. Our approach is thus named *Max-Min Sliced Gromov Wasserstein* (MSGW). Our method appears to be the first sliced GW distance that preserves the rotation and reflection invariance of the GW distance, while remaining computationally feasible. A comparison of our proposed method with existing works is summarized in Table 1.

**Related work**  Our work is most closely related to the sliced Gromov-Wasserstein distance introduced in Vayer et al. (2019b). Therein, the authors proposed a naive sliced GW (SGW) distance, which loses the rotation and reflection invariance of the GW distance. To address this, they also introduced a rotational invariant SGW (RISGW), which however requires optimizing over the Stiefel

| Distance | Rotation & reflection invariant | Complexity[1] | Empirical optimality |
|---|---|---|---|
| GW | Yes | NP-hard; $\mathcal{O}(n^4)$; $\mathcal{O}(n^3)$ | good |
| Entropic GW | Yes | $\mathcal{O}(n^4)$; $\mathcal{O}(n^3)$ | good |
| SGW | No | $\mathcal{O}(Ln\log(n))$ | good |
| RISGW | Yes | $\mathcal{O}(n_{\text{iter}}(Ln(p+\log(n))+p^3))$ | not reliable enough |
| **MSGW (ours)** | Yes | $\mathcal{O}(L^2 n\log(n))$ | good |

Table 1: Characteristics of several sliced Gromov-Wasserstein distances. The original GW distance is NP-hard and prohibitively expensive to compute. For local convergence, naive implementation gives $\mathcal{O}(n^4)$. Another widely-adopted method introduced in Peyré et al. (2016) further reduces complexity to $\mathcal{O}(n^3)$ for decomposable loss functions, for both GW and entropic GW. Previously proposed sliced versions in Vayer et al. (2019b) either lose the rotation and reflection invariance of GW (SGW), or are numerically expensive and unstable (RISGW). Our proposed method MSGW preserves the invariance properties of GW while being computationally affordable and stable.

manifold and therefore introduces a huge computational cost in higher dimensions. The paper builds on a line of research that leverages the closed-form solution of the one-dimensional Wasserstein problem, which have first been proposed in Julien et al. (2011); Bonneel et al. (2015); Paty & Cuturi (2019). Several variants of sliced methods include Max-Sliced Wasserstein Deshpande et al. (2019), Generalized Sliced Wasserstein Kolouri et al. (2019b), Spherical Sliced Wasserstein Bonet et al. (2022), Hierarchical Sliced Wasserstein Nguyen et al. (2022), and Energy-Based Sliced Wasserstein Nguyen & Ho (2023). The corresponding applications include generative models Wu et al. (2019); Deshpande et al. (2019); Kolouri et al. (2018); Bunne et al. (2019), image processing Nguyen & Ho (2024); Julien et al. (2011), and kernel methods Kolouri et al. (2016). Slicing techniques have also been applied to general probability divergences, including Sliced Mutual Information Goldfeld & Greenewald (2021); Fayad & Ibrahim (2023), Sliced-Cramér Kolouri et al. (2019a), and Max-Sliced Mutual Information Tsur et al. (2023). The statistical and topological properties of general divergences are explored in (Nadjahi et al., 2020). A similar approach to slicing is the "linear" OT, where distances are computed by projecting the manifold onto the tangent plane. This category includes Linear Wasserstein Wang et al. (2013), Linear GW Beier et al. (2022), and Linear Fused GW Nguyen & Tsuda (2023).

Recent works have considered globally solving the GW problem and Gromov-Hausdorff problem, but this requires special conditions or relaxations Villar et al. (2016); Mula & Nouy (2024); Ryner et al. (2023); Chen et al. (2024); Dumont et al. (2025). Thus, various other methods have been proposed to approximate the solution of GW, thus reducing the computational burden. An original approach is to use entropic regularization Gold & Rangarajan (1995); Solomon et al. (2016) and the Sinkhorn algorithm Cuturi (2013) celebrated for OT problems. Recent work Zhang et al. (2024) further reduces the time complexity of this entropic problem by fast gradient computation. Other works focus on the computational bottleneck caused by the cost matrices, specifically the tensor-matrix product. That is, Peyré et al. (2016) introduced a technique for decomposable loss functions. For more general cases, a iterative sampled method is introduced in Kerdoncuff et al. (2021); low-rank approaches were introduced for the cost matrices and couplings in Scetbon et al. (2022); a randomized sparsification method for the cost matrices was proposed in Li et al. (2023).

**Contributions**  We introduce MSGW - a max-min sliced Gromov Wasserstein distance. Our contributions are detailed as follows:

- We define the novel MSGW distance, which improves on the existing sliced GW distance by combining it with a max-min formulation. The MSGW formulation can be seen as two adversaries playing a game to find one-dimensional projection directions (that is, slices) that yield the maximum discrepancy between the associated one-dimensional probability distributions. This approach thus has a computational cost similar to that of the naive SGW distance and is significantly cheaper than its adaptation RISGW.
- We prove that MSGW preserves the metric properties of the original GW distance (Theorem 3.3), and MSGW is a metric up to measure-preserving isometries, i.e., it induces the same equivalence relation as GW on metric measure spaces.

---

[1]$n$ = number of data points; $L$ = number of projection directions; $p$ = dimension of the problem; $n_{iter}$ = number of iterations for internal optimization method.

- We show that MSGW can be seen as a pseudo Hausdorff distance (Theorem 3.4). Based on this observation we prove an error bound for the practical case, where only a finite set of projection directions can be computed (Proposition 4.2).
- We experimentally validate the competitive performance of MSGW, resulting from its isometric properties and computational advantages, on various point cloud datasets and as a loss function in a generative adversarial network (GAN).

To the best of our knowledge, *MSGW is the first sliced formulation of the GW distance that preserves the invariance of the GW distance to rotations and reflections while being computationally efficient*, see Table 1.

**Notations**  We denote by $\mathbb{P}(\mathbb{R}^p)$ the space of probability measures on $\mathbb{R}^p$, and let $\|\cdot\|_{2,p}$ denote the Euclidean distance in $\mathbb{R}^p$. We often omit the subindex $p$ when the space is clear from the context. The $p$-dimensional hypersphere is denoted as $\mathbb{S}^{p-1} = \{\theta \in \mathbb{R}^p : \|\theta\|_2 = 1\}$. For $\theta \in \mathbb{S}^{q-1}$ denote $P_\theta : \mathbb{R}^q \to \mathbb{R}$ the projection in direction $\theta$, i.e., $P_\theta(x) = \langle \theta, x \rangle$. The delta measure $\delta_{\boldsymbol{x'}} \in \mathbb{P}(\mathbb{R}^p)$ for a point $\boldsymbol{x'} \in \mathbb{R}^p$ is defined as $\delta_{\boldsymbol{x'}}(\boldsymbol{x}) = 1$ if $\boldsymbol{x} = \boldsymbol{x'}$, and $\delta_{\boldsymbol{x'}}(\boldsymbol{x}) = 0$ otherwise. The push-forward operator for a continuous function $f$ is denoted as $f\#$. Note that the push-forward of the delta function is given by $f\#\delta_{\boldsymbol{x}} = \delta_{f(\boldsymbol{x})}$.

## 2 BACKGROUND AND PREVIOUS WORKS

In this Section, we introduce the Gromov-Wasserstein distance and discuss existing computational methods for solving it. In particular, we present the sliced Gromov-Wasserstein method with its variations and their limitations. In the present work, we propose a new variant of the sliced Gromov-Wasserstein distance, which addresses these limitations.

**Gromov-Wasserstein distance**  In this work, we consider the discrete GW distance, where the structured data is given by discrete measures $\mu = \sum_{i=1}^n \boldsymbol{p}_i \delta_{\boldsymbol{x}^{(i)}}$ and $\nu = \sum_{j=1}^m \boldsymbol{q}_j \delta_{\boldsymbol{y}^{(j)}}$ that lie on two possibly different Euclidean spaces $\mathbb{R}^p$ and $\mathbb{R}^q$. Here $\boldsymbol{x}^{(i)} \in \mathbb{R}^p$ and $\boldsymbol{y}^{(j)} \in \mathbb{R}^q$ are the support points of the two measures, and $\boldsymbol{p} \in \mathbb{R}^n$ and $\boldsymbol{q} \in \mathbb{R}^m$ are probability vectors that describe the mass on each of the support points. The structure of the measures $\mu$ and $\nu$ is captured in two matrices $\boldsymbol{C}^\mu \in \mathbb{R}^{n \times n}$ and $\boldsymbol{C}^\nu \in \mathbb{R}^{m \times m}$, where the elements are defined as $\boldsymbol{C}^\mu_{i,i'} = \|\boldsymbol{x}^{(i)} - \boldsymbol{x}^{(i')}\|_{2,p}^2$ and $\boldsymbol{C}^\nu_{j,j'} = \|\boldsymbol{y}^{(j)} - \boldsymbol{y}^{(j')}\|_{2,q}^2$. This gives rise to the GW distance

$$GW(\mu, \nu) \stackrel{\text{def.}}{=} \min_{\boldsymbol{T} \in \mathcal{T}(\boldsymbol{p},\boldsymbol{q})} \sum_{i,i',j,j'} \left| \boldsymbol{C}^X_{i,i'} - \boldsymbol{C}^Y_{j,j'} \right|^2 \boldsymbol{T}_{i,j} \boldsymbol{T}_{i',j'}, \tag{1}$$

where $\mathcal{T}(\boldsymbol{p}, \boldsymbol{q}) = \{\boldsymbol{T} \in \mathbb{R}^{n \times m} : \sum_{j=1}^m \boldsymbol{T}_{ij} = \boldsymbol{p}_i, \sum_{i=1}^n \boldsymbol{T}_{ij} = \boldsymbol{q}_j\}$ is the set of all possible couplings between the probability vectors. Thus, the product $\boldsymbol{T}_{i,j}\boldsymbol{T}_{i',j'}$ represents the joint probability of matching two pairs of points $(\boldsymbol{x}^{(i)}, \boldsymbol{y}^{(j)})$ and $(\boldsymbol{x}^{(i')}, \boldsymbol{y}^{(j')})$ (Vayer, 2020). Note that the structure matrices $\boldsymbol{C}^\mu$ and $\boldsymbol{C}^\nu$ indicate the *similarity* of pairs of support points within each measure. Hence, the GW distance measures the *distortion* of the *similarity* between each pair of support points in the two measures (Chowdhury & Needham, 2020).

Mathematically, the GW distance is interesting for comparing structured data, because it is a metric on the space of probability distributions, up to measure-preserving isometries. More precisely, GW is non-negative, symmetric, and satisfies the triangle equation. Moreover, $GW(\mu, \nu) = 0$ if and only if the measures $\mu$ and $\nu$ are isomorphic in the following sense, cf. Vayer et al. (2019b).

**Definition 2.1** (Measure-preserving isometry). *Two measures on Euclidean spaces $\mu \in \mathbb{P}(\mathbb{R}^p)$ and $\nu \in \mathbb{P}(\mathbb{R}^q)$ are called isomorphic if there exists a measure-preserving isometry between them, that is, a bijective mapping $f : \mathbb{R}^p \to \mathbb{R}^q$ such that*

1. *(f is measure preserving) $f_\#\mu = \nu$,*
2. *(f is an isometry) for all $x, x' \in \mathbb{R}^p$ it holds that $\|x - x'\|_{2,p} = \|f(x) - f(x')\|_{2,q}$.*

Unfortunately, the quadratic optimization problem in equation 1 is non-convex and therefore in general NP hard (Vayer, 2020). A global minimum can only be found efficiently in special cases (Ryner et al., 2023; Dumont et al., 2025).

**Sliced Gromov-Wasserstein distance**  Inspired by the popular sliced Wasserstein distance (Julien et al., 2011), where high-dimensional measures are projected on a one-dimensional space, in which

the computation of the Wasserstein distance is significantly cheaper, Vayer et al. (2019b) proposed to approximate the GW distance in equation 1 by a sliced Gromov-Wasserstein (SGW) distance. Without loss of generality, we here assume that $p \leq q$. Let $\Delta : \mathbb{R}^p \to \mathbb{R}^q$ be a function that maps from the lower to the higher-dimensional space, and let $\mu_\Delta = \Delta_{\#}\mu \in \mathbb{P}(\mathbb{R}^q)$ be the measure mapped to that space[2]. Then the SGW distance is defined as

$$SGW_\Delta(\mu, \nu) = \mathop{\mathbb{E}}_{\theta \sim \mathrm{unif}(\mathbb{S}^{q-1})} \left[ GW\left((P_\theta)_{\#}\mu_\Delta, (P_\theta)_{\#}\nu\right) \right]. \tag{2}$$

This distance can be computed at a significantly lower cost than the GW distance, as it only requires solving GW problems (equation 1) of measures in $\mathbb{P}(\mathbb{R})$. If the measures have the same number of support points $n = m$ with uniform weights $\boldsymbol{p}_i = \boldsymbol{q}_j = 1/n$, the optimization problem can often be solved at the cost of a sorting algorithm, i.e., $\mathcal{O}(n \log(n))$. For more details, we refer the reader to (Vayer et al., 2019b, Section 3) and Appendix B. Although follow-up works (Beinert et al., 2022; Dumont et al., 2025) showed that counter-examples to this observation exist, the sorting approach in (Vayer et al., 2019b) typically works well in practice and is the standard method used in the field[3].

However, a disadvantage with SGW is that it is not rotation and reflection invariant. To see this, consider the setting where $p = q$. If $\nu$ is a rotated (or reflected) version of $\mu$, SGW is only zero if $\Delta$ is chosen to be exactly this rotation. However, since the rotation is unknown, a canonical choice in this setting is typically the identity map. The issue becomes even worse when $p \neq q$ and $\Delta$ is a zero-padding function, as this introduces additional bias in equation 2.

**Rotation Invariant Sliced Gromov-Wasserstein distance** To remedy the fact that the SGW distance is not rotation invariant, Vayer et al. (2019b) introduced the Rotation Invariant SGW (RISGW) distance, which also optimizes over the function $\Delta$ in equation 2. More precisely, they allow for all choices of $\Delta$ that are linear isometries. These can be described as the set of matrices in $\mathbb{R}^{q \times p}$ with orthonormal columns, i.e., $\Delta \in \mathbb{V}_p(\mathbb{R}^q) = \left\{ \Psi \in \mathbb{R}^{q \times p} \mid \Psi^T \Psi = I_p \right\}$, also called the Stiefel manifold. The RISGW distance reads thus

$$RISGW(\mu, \nu) = \min_{\Psi \in \mathbb{V}_p(\mathbb{R}^q)} SGW^\Psi(\mu, \nu). \tag{3}$$

In contrast to the SGW distance, the RISGW distance is therefore invariant to rotations and reflections. However, it requires optimizing over the Stiefel manifold, which is typically done by a gradient descent over the manifold. Assuming the gradient descent requires $n_{iter}$ iterations until convergence, the total complexity is $\mathcal{O}\left(n_{iter}(Ln(p + \log(n)) + p^3)\right)$ Vayer et al. (2019b). Unfortunately, this is prohibitively expensive in high-dimensional problems.

## 3 MAX-MIN SLICED GROMOV-WASSERSTEIN DISTANCE

As discussed in Section 2 and summarized in Table 1, existing sliced Gromov-Wasserstein distances either lose the isometric invariance of the GW distance, or come with a prohibitively high computational cost. These limitations call for a novel sliced GW method that addresses both these limitations, which we will introduce in this section.

**Definition 3.1** (Max-min sliced Gromov-Wasserstein distance). *For measures $\mu \in \mathbb{P}(\mathbb{R}^p)$ and $\nu \in \mathbb{P}(\mathbb{R}^q)$, we define the max-min sliced Gromov-Wasserstein distance as*

$$MSGW(\mu, \nu) =$$

$$\max \left\{ \sup_{\theta \in \mathbb{S}^{p-1}} \inf_{\phi \in \mathbb{S}^{q-1}} GW\left((P_\theta)_{\#}\mu, (P_\phi)_{\#}\nu\right), \sup_{\phi \in \mathbb{S}^{q-1}} \inf_{\theta \in \mathbb{S}^{p-1}} GW\left((P_\theta)_{\#}\mu, (P_\phi)_{\#}\nu\right) \right\}. \tag{4}$$

Similar to SGW, our proposed distance exploits the computational advantage of comparing only one-dimensional projections of the given measures. However, instead of projecting both measures into the same direction, we instead allow for them to be projected onto different directions, and thus maintain the rotation invariance of the GW distance. Compared with the RISGW distance, MSGW optimizes directly over the projection directions and therefore circumvents the expensive optimization over the Stiefel manifold.

---

[2]For instance, $\Delta$ can be a "zero-padding" function that adds zeros to the support of $\mu$ such that it becomes the same dimension as $\nu$. If $p = q$ we can choose $\Delta$ to be the identity map.

[3]Thus, we also utilize this method for our novel sliced GW distance, see Section 4

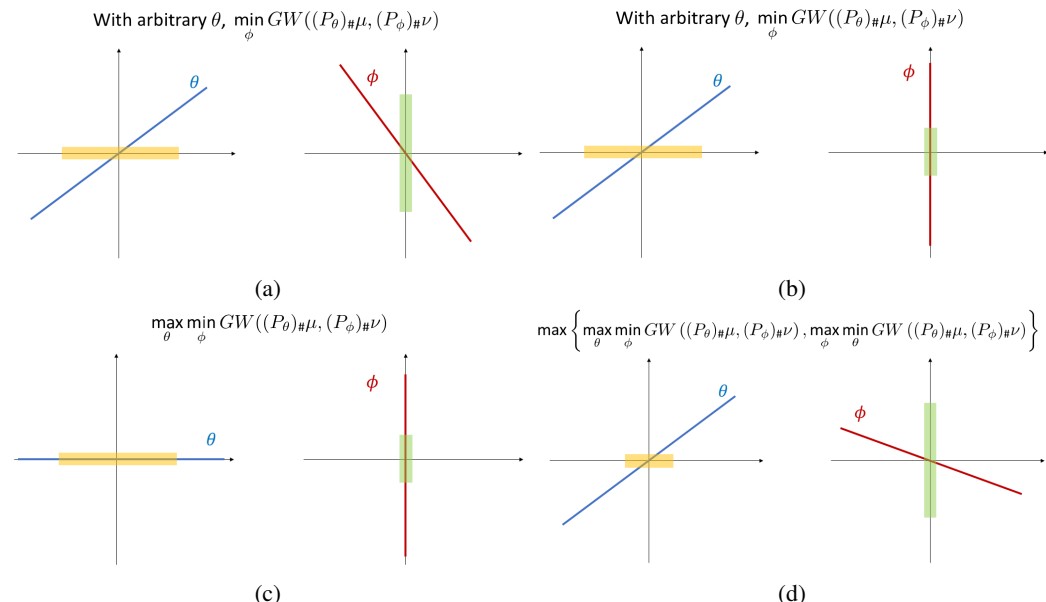

Figure 1: **Intuitive illustration of MSGW.** The supports of two measures $\mu$ and $\nu$ are shown in orange shade and green shade, respectively. For simplicity of illustration, the supports are all rectangular, with possibly different lengths. The blue line and red line show the projection directions $\theta$ and $\phi$. (a) The two measures are isometric with a rotation in difference. We can easily see that with an arbitrary $\theta$, we can always find a corresponding $\phi$ such that $GW((P_\theta)_\#\mu, (P_\phi)_\#\nu) = 0$. (b) Now we make the support of $\nu$ shorter, so the two measures are not isometric anymore. For certain $\theta$ (e.g., if we pick the direction as shown in the figure), we can still find a corresponding $\phi$ such that $GW((P_\theta)_\#\mu, (P_\phi)_\#\nu) = 0$. This projection setting is not suitable for distinguishing the two measures, since it does not show the important difference in the "length" of supports. (c) We want to choose two projections that can distinguish the two supports maximally, as shown in this subfigure. This can be achieved with a max-min formulation $\max_\theta \min_\phi GW((P_\theta)_\#\mu, (P_\phi)_\#\nu)$. (d) We note that there is a "bias" or "asymmetry" in the single max-min formulation - it only works if the support of $\nu$ is shorter than the support of $\mu$. To remedy this, we symmetrize the problem by maximizing over two max-min formulations, which yields the final form of MSGW.

Intuitively, MSGW searches for each projection direction $\theta \in \mathbb{S}^{p-1}$ the "best" direction $\phi \in \mathbb{S}^{q-1}$ such that the projected measures $(P_\theta)_\#\mu$ and $(P_\phi)_\#\nu$ become as similar as possible, as measured by the one-dimensional GW distance. These terms can thus be seen as two actors "playing a game", where the inf-term aims to preserve isometry and the sup-term represents an adversary who finds the worst scenario, in order to distinguish different objects. The outer maximization in Defintion 3.1 guarantees that MSGW is symmetric in the two measures. This procedure is illustrated in Figure 1.

**Theorem 3.2** (Dual property between MSGW and GW.). *For any $\mu \in \mathbb{P}(\mathbb{R}^p)$ and $\nu \in \mathbb{P}(\mathbb{R}^q)$, it holds that*

$$MSGW(\mu, \nu) = 0 \iff GW(\mu, \nu) = 0. \tag{5}$$

**Theorem 3.3** (MSGW is a metric up to measure-preserving isometries.). *MSGW is a metric up to measure-preserving isometries, i.e., for any $\mu \in \mathbb{P}(\mathbb{R}^p)$, $\nu \in \mathbb{P}(\mathbb{R}^q)$, and $\gamma \in \mathbb{P}(\mathbb{R}^r)$ it holds that*

1. *(Positivity) $MSGW(\mu, \nu) \geq 0$, with equality if and only if $\mu$ and $\nu$ are isomorphic[4].*
2. *(Symmetry) $MSGW(\mu, \nu) = MSGW(\nu, \mu)$.*
3. *(Triangle inequality) $MSGW(\mu, \nu) \leq MSGW(\mu, \gamma) + MSGW(\gamma, \nu)$.*

Theorems 3.2 and 3.3 indicate that MSGW satisfies the same metric properties as the original GW, including translation, rotation, and reflection invariance, which are *not* inherited by SGW.

---

[4]"isomorphic" means "measure-preserving isometric". More formally, two metric measure spaces $(X, d_X, \mu)$ and $(Y, d_Y, \nu)$ are isomorphic if and only if there exists a measure-preserving isometry $f : X \to Y$ between them (Mémoli, 2011; Vayer et al., 2018).

It is worth noting that MSGW has a similar structure as the Hausdorff distance, which defines a metric between two sets $A$ and $B$ within the same metric space $(Z, d)$ as

$$d_{\mathcal{H}}^{Z}(A, B) := \max \left( \sup_{a \in A} \inf_{b \in B} d(a, b), \sup_{b \in B} \inf_{a \in A} d(a, b) \right). \tag{6}$$

In fact, MSGW can be formulated as a pseudo Hausdorff distance between sets of one-dimensional measures, as follows.

**Theorem 3.4** (MSGW is equivalent to a pseudo Hausdorff distance)**.** *Define the two sets consisting of one-dimensional measures*

$$\boldsymbol{\mu}^{\mathbb{S}^{p-1}} = \{(P_\theta)_{\#}\mu \mid \theta \in \mathbb{S}^{p-1}\} \subset \mathbb{P}(\mathbb{R}), \qquad \boldsymbol{\nu}^{\mathbb{S}^{q-1}} = \{(P_\phi)_{\#}\nu \mid \phi \in \mathbb{S}^{q-1}\} \subset \mathbb{P}(\mathbb{R}). \tag{7}$$

*Then it holds that*

$$MSGW(\mu, \nu) = d_{\mathcal{H}}^{\mathbb{P}(\mathbb{R})}(\boldsymbol{\mu}^{\mathbb{S}^{p-1}}, \boldsymbol{\nu}^{\mathbb{S}^{q-1}}),$$

*where $d_{\mathcal{H}}^{\mathbb{P}(\mathbb{R})}$ denotes the Hausdorff distance defined in equation 6 on the pseudo-metric space $(\mathbb{P}(\mathbb{R}), GW)$.*

# 4 COMPUTATIONAL APPROXIMATION OF MSGW DISTANCE

Evaluating the MSGW distance exactly would require solving the nested sup-inf problems in equation 4 over the unit spheres. In order to compute the MSGW distance in practice, we solve the problems over discrete direction sets $\Theta \subset \mathbb{S}^{p-1}$ and $\Phi \subset \mathbb{S}^{q-1}$. We denote this finite direction-sample approximation of MSGW as

$$MSGW_{\Theta,\Phi}(\mu, \nu) = \max \left\{ \max_{\theta \in \Theta} \min_{\phi \in \Phi} GW\left((P_\theta)_{\#}\mu, (P_\phi)_{\#}\nu\right), \max_{\phi \in \Phi} \min_{\theta \in \Theta} GW\left((P_\theta)_{\#}\mu, (P_\phi)_{\#}\nu\right) \right\}. \tag{8}$$

**Remark 4.1** (Complexity)**.** *The one-dimensional GW distances in equation 8 can typically be solved by an ordering algorithm, as described in Appendix B. These have complexity $\mathcal{O}(n \log(n))$, where $n$ is the number of support points of $\mu$ and $\nu$. If we sample $L$ projection directions in both spaces, i.e., $|\Theta| = |\Phi| = L$, then $L^2$ one-dimensional GW distances must be compared to solve the max-min problems in equation 8. The total complexity of computing MSGW is thus $\mathcal{O}(L^2 n \log(n))$. This is slightly higher than the naive SGW distance in equation 2, which has complexity $\mathcal{O}(Ln \log(n))$, but preserves the metric properties of the GW distance. Moreover, note that the complexity of our method is significantly lower than the previous rotational invariant method RISGW in equation 3. In Section 5, we will also confirm empirically that the computation time of RISGW is often much higher than that of MSGW.*

The error incurred by using the finite sample approximation of MSGW depends on how well the 1D projections on the unit spheres, defined in equation 7, are approximated by their discrete counterparts

$$\boldsymbol{\mu}^{\Theta} = \{(P_\theta)_{\#}\mu \mid \theta \in \Theta\} \subset \mathbb{P}(\mathbb{R}), \qquad \boldsymbol{\nu}^{\Phi} = \{(P_\phi)_{\#}\nu \mid \phi \in \Phi\} \subset \mathbb{P}(\mathbb{R}). \tag{9}$$

More precisely, we get the following error bound.

**Proposition 4.2.** *For measures $\mu \in \mathbb{P}(\mathbb{R}^p)$ and $\nu \in \mathbb{P}(\mathbb{R}^q)$ it holds*

$$\left| MSGW(\boldsymbol{\mu}^{\mathbb{S}^{p-1}}, \boldsymbol{\nu}^{\mathbb{S}^{q-1}}) - MSGW(\boldsymbol{\mu}^{\Theta}, \boldsymbol{\nu}^{\Phi}) \right| \leq MSGW(\boldsymbol{\mu}^{\mathbb{S}^{p-1}}, \boldsymbol{\mu}^{\Theta}) + MSGW(\boldsymbol{\nu}^{\mathbb{S}^{q-1}}, \boldsymbol{\nu}^{\Phi}) \tag{10}$$

# 5 EXPERIMENTS

This section presents experiments that validate the properties of MSGW, compare them with SGW and RISGW, and illustrate its use in GANs.

**Translation, rotation and reflection** invariance    We first performed experiments on the spiral datasets shown in Vayer et al. (2019b) and verified the invariance of MSGW under translation and reflection. Since a rotation can be viewed as a composition of multiple reflections, reflection invariance automatically implies rotation invariance. Figure 2 shows the values of GW, SGW, MSGW, and RISGW with respect to the different reflection angles, i.e., the angle of the reflection line through

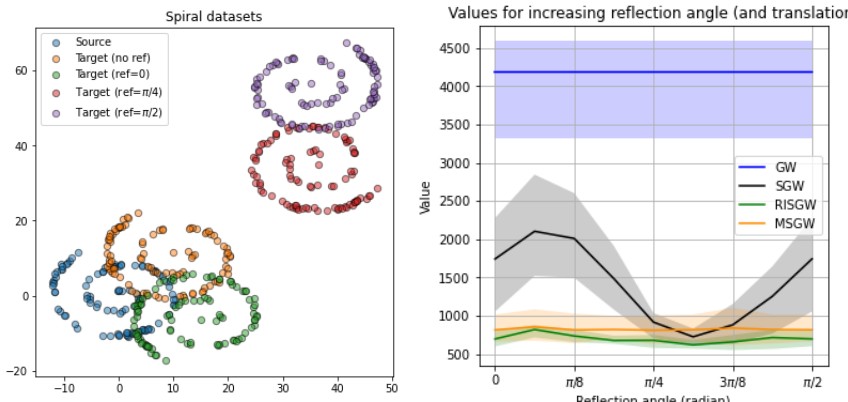

Figure 2: **Translation, rotation and reflection invariance:** Using MSGW, SGW, RISGW, and GW to compare spiral datasets with different reflection angles with 5 trials. (Left) The plots of the spiral distributions in the first trial, including the source and the targets with different reflections. (Right) Distance values of MSGW, SGW, GW, and RISGW with $n = 100$ samples and $L = 20$ projection directions regarding the reflection angle of the target. The maximum number of optimization iterations of GW and RISGW is set to 600. The shaded area indicates the 20%-80% percentiles.

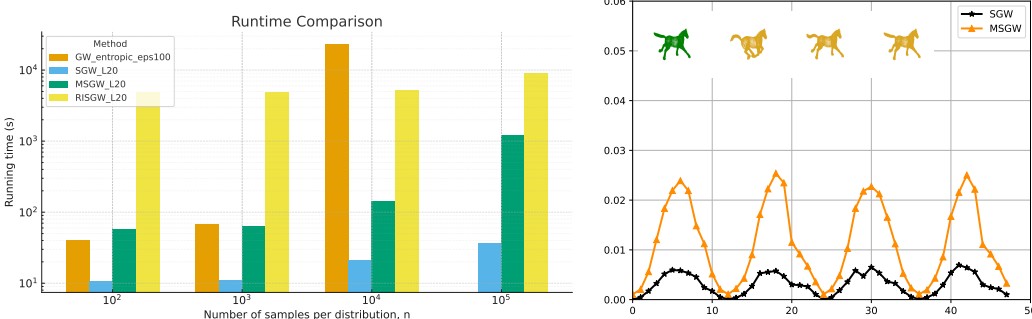

Figure 3: Runtime comparison between GW, SGW, RISGW, and the proposed MSGW.

Figure 4: **Meshes:** Using SGW and MSGW to compare the mesh data.

the origin with the x-axis. Targets are also added with arbitrary translations. Empirically, RISGW is less stable than MSGW in general, i.e., the curve is not always flat, though we pick a relatively flat result to show here with random seed set as 12. This is likely due to the difficulty in optimizing its non-convex objective on the Stiefel manifold. The results with different random seeds and different $L$ are shown in Appendix D.1 in Figure 11.

**Runtime comparison** Figure 3 compares the runtime of the proposed MSGW with entropic GW (with $\epsilon = 100$), SGW, and RISGW with different numbers of samples $n$, all of which were implemented in PyTorch and run on an RTX 2080Ti GPU. The number of projection directions for SGW, MSGW, and RISGW is chosen as $L = 20$. The result of entropic GW is not available when $n = 10^5$ as it causes memory overflow. It can be observed that the runtime of MSGW is modest compared to that of RISGW.

**Meshes** We used SGW and MSGW on the horse mesh data set in Vayer et al. (2019b) and show the difference between these two distances. The dataset consists of 48 horse meshes with cyclical motions. Each horse is constructed by $n = 8431$ sampled points of 3 dimensions. We use $L = 500$ projection directions for both SGW and MSGW. Figure 4 shows the distances between the source horse (the green one) and the 48 target horses (we show three examples in yellow). We also observe the numerical error when we compute the SGW and MSGW distances with finite numbers of projections, especially when we compute the distance between two identical horses (the bottoms of the two plots). The MSGW shows slightly greater numerical errors than SGW, possibly due to the difficulty in finding the *two* exact projection directions that lead to the zero distance, instead of *one* in SGW.

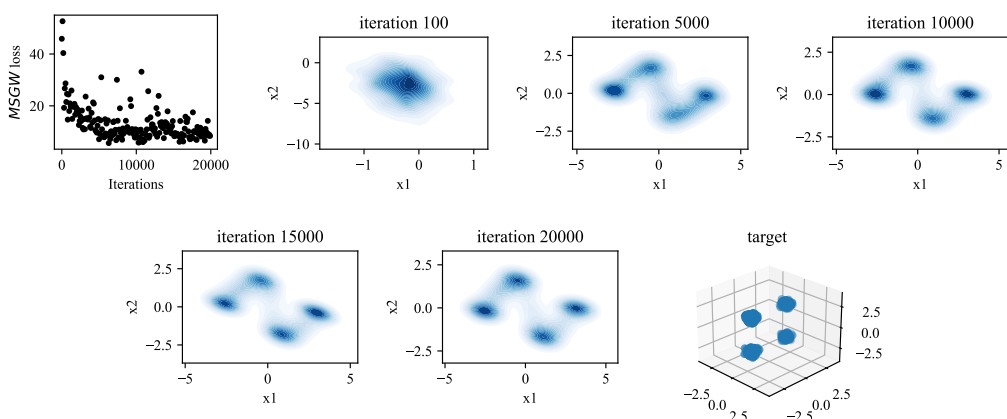

Figure 5: **MSGW GAN:** Using MSGW distance as the loss function of the generator in GAN with $L = 20$. (1st) The loss value evolution regarding iteration. (2nd to 6th) The generated datasets of different iterations. (7th or last) The target 3D-4mode Gaussian dataset.

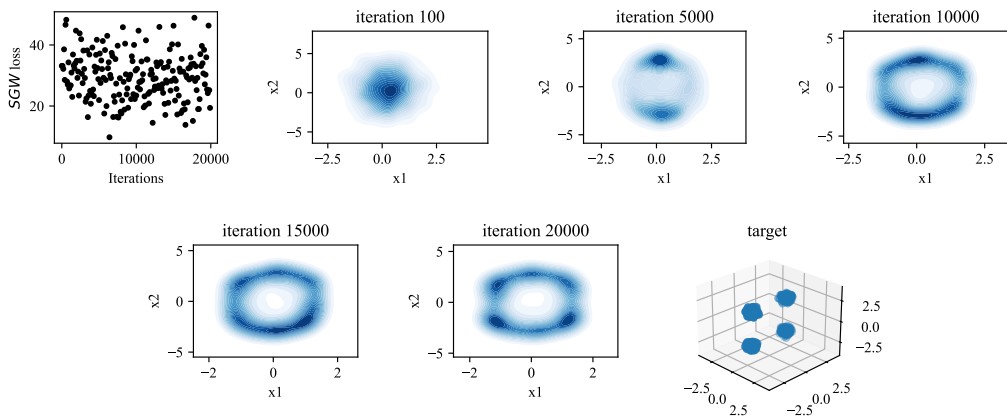

Figure 6: **SGW GAN:** Using the same experiment settings as Figure 5, but use SGW instead of MSGW.

**GAN** The GW and SGW distances have been proposed to act as a loss function in the generator of a generative adversarial network (GAN) (Bunne et al., 2019; Vayer et al., 2019b) to compare across different spaces. Following the experiments conducted in (Vayer et al., 2019b), the following experiments are conducted for MSGW loss accordingly without training the adversary, and the number of projection directions as $L = 20$. The target dataset is constructed as 3D-4mode Gaussian, i.e., 4 clusters of Gaussian points, each cluster centered around a fixed center. We also maintain the same experimental settings used in Bunne et al. (2019), i.e., the generator is a multi-layer perceptron constructed with 3 hidden layers, each of 128 neurons with ReLU activation functions, with the input layer of 256 neurons and output layer of 2 neurons. We set the number of sample points for training as 40,000, batch size as 256, and use 1,000 data points for plotting. The latent dimension of the generator is set as 256. We use the Adam optimizer with a learning rate of $2 \times 10^{-4}$ and $\beta_1 = 0.5, \beta_2 = 0.99$.

Figure 5 shows the evolution of loss values and the generated dataset regarding the number of iterations with a maximum of 20,000. The generator is able to generate satisfying datasets after 5000 iterations, and the loss value becomes relatively stable after 10,000 iterations. The results of the same experiment settings using SGW instead of MSGW are shown in Figure 6. We can observe that SGW performs poorly with such settings. Experiments conducted with different numbers of projection directions $L$ and target as a 4D-4mode Gaussian mixture are shown in Appendix D.2.

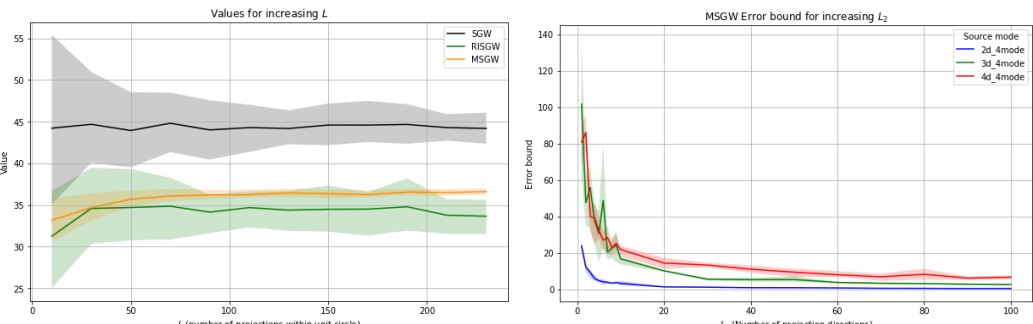

Figure 7: **Number of projections and error bound:** Illustration of SGW, RISGW, and MSGW on Gaussian datasets for varying number of projections. The shaded area indicates the 20%-80% percentiles of distance values. (Left) The SGW, RISGW, and MSGW distance values regarding the increasing number of projections (with 50 trials). (Right) Error bounds of MSGW distance with 5 trials, with target set as 2D-4mode and source set as {2D-4mode,3D-4mode,4D-4mode}.

**Number of projections**  To investigate the influence of the increasing number of projections, for SGW, RISGW, and MSGW, we use the 2D-4mode and 3D-4mode Gaussian mixtures as the source and target. Each trial uses a different set of projection directions. Figure 7 (Left) indicates that, compared with SGW and RISGW, MSGW exhibits greater stability with random projection samples and converges more quickly as the number of projections increases.

**Error bound**  We compute the error bound described on the RHS of equation 10. Since we do not have access to the ground truth for the infinite sets $\boldsymbol{\mu}^{\mathbb{S}^{p-1}}$ and $\boldsymbol{\mu}^{\mathbb{S}^{p-1}}$, we approximate the continuous unit spheres $\mathbb{S}^{p-1}$ and $\mathbb{S}^{q-1}$, by $L_1 = 1000$ projection directions. Moreover, we set $L_2 = |\Theta| = |\Phi| = \{1, 2...10, 20, ..., 100\}$. Figure 7 (Right) shows the evolution of the MSGW error bound as a function of the size of the discrete sets, $L_2$, for various source modes. The source is set to {2D-4mode,3D-4mode,4D-4mode} Gaussian mixtures, and the target is always set to be 2D-4mode Gaussian mixtures. The sample size of all datasets is $n = 256$. The lines are plotted with 5 trials. The figure shows that all error bounds drop significantly around $L_2 = 10$, indicating that already a small number of projections can yield to a promising approximation of MSGW. We also notice that the higher-dimensional dataset has higher error bounds and experiences more fluctuations.

**Robustness**  Following the experiments shown in Paty & Cuturi (2019), we conduct the experiments on how noise affects the distances. The source and target are randomly generated with 256 samples in each trial. We test the case when the source $\mu_0$ and the target $\nu_0$ are respectively 2D-Gaussian and 3D-Gaussian distributions. By adding different levels of zero-mean Gaussian noise $\sigma \mathcal{N}(0, I)$ to the original dataset, we have new datasets $\mu_\sigma$ and $\nu_\sigma$, then compute the relative error with respect to $\sigma$ as

$$\sigma \mapsto \frac{|\, d\,(\mu_\sigma, \nu_\sigma) - d\,(\mu_0, \nu_0)\,|}{d\,(\mu_0, \nu_0)}$$

where $d$ can be SGW, GW, RISGW, and MSGW. The results with $L = 20$ and 50 trials are shown in Figure 8. We can see that for high noise, MSGW is the most robust method out of the tested ones.

## 6  CONCLUSION AND FUTURE WORK

We introduced the MSGW distance, the first sliced formulation of GW that preserves invariance to rotations and reflections while remaining computationally efficient (to the best of our knowledge). Experiments confirm that MSGW achieves strong performance with few projections and scales well across dimensions, establishing it as a practical and robust alternative for cross-metric space comparisons.

Future research may include how different factors in various applications will affect the performance of MSGW, such as the construction of a multilayer perception and optimizer in GAN. More properties of MSGW can also be explored, including its further connections to the Hausdorff distance, GW

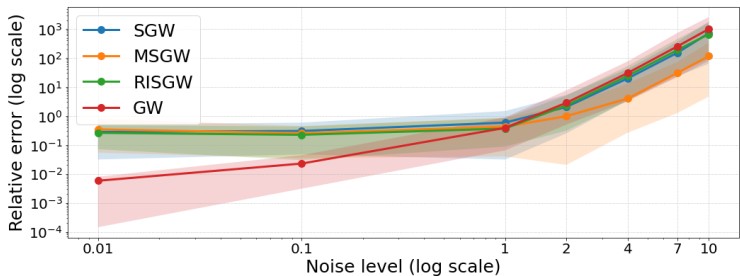

Figure 8: **Relative error in noisy cases:** Relative error of GW, SGW, RISGW and MSGW with different noise levels shown in log-log scale (with 50 trials). The shaded areas represent the 10%-90% percentiles.

distance, and the $\infty$-Wasserstein distance. It may also be beneficial to adopt continuous optimization regarding $\theta$ and $\phi$ in the max-min formulation in certain applications.

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

# Supplementary material for
# Max-Min Sliced Gromov-Wasserstein

Here, we provide supplementary material for the submission titled "Max-Min Sliced Gromov-Wasserstein". The supplementary material is structured as follows: In Section B we provide technical details on the Gromov-Wasserstein distance in one-dimensional space. This is a basic ingredient in all sliced Gromov-Wasserstein methods. In Section C we prove the theoretical results stated in the main paper. Finally, Section D.2 contains additional experimental results.

## A    THE USE OF LARGE LANGUAGE MODELS (LLMS)

LLMs have been used for partial code generation and text revision.

## B    GROMOV-WASSERSTEIN IN ONE DIMENSION

In this section, we present a summary of the sorting method for solving 1D GW, originally shown in Vayer et al. (2019b), and its counterexamples. We also follow a new result that shows that the sorting method is *indeed* true when $GW = 0$.

### B.1    THE SORTING METHOD FOR 1D GW

The idea of sliced Gromov-Wasserstein distances is to project the given measures onto one-dimensional lines, where the optimization problem becomes significantly cheaper to solve. In particular, we consider the case where the measures $\mu$ and $\nu$ have the same number of support points $n = m$ with uniform weights $\boldsymbol{p}_i = \boldsymbol{q}_i = 1/n$. In this case, the solution $\boldsymbol{T}$ to the optimization problem in equation 1 is a permutation matrix. Then, equation 1 can be formulated as the so-called Gromov-Monge problem

$$\min_{\sigma \in \text{perm}\{1,\dots,n\}} \frac{1}{n^2} \sum_{i,j} |\boldsymbol{C}_{i,j}^X - \boldsymbol{C}_{\sigma(i),\sigma(j)}^Y|^2. \tag{11}$$

If we let $\boldsymbol{C}_{i,j}^X = |x_i - x_j|^\alpha$ and $\boldsymbol{C}_{\sigma(i),\sigma(j)}^Y = |y_{\sigma(i)} - y_{\sigma(j)}|^\alpha$ with $\alpha > 0$, equation 11 is equivalent to

$$\max_{\sigma \in \text{perm}\{1,\dots,n\}} \frac{1}{n^2} \sum_{i,j} |x_i - x_j|^\alpha \cdot |y_{\sigma(i)} - y_{\sigma(j)}|^\alpha \tag{12}$$

If additionally the support points of the two distributions, $x_1, \dots, x_n \in \mathbb{R}$ and $y_1, \dots, y_n \in \mathbb{R}$ are ordered, i.e., $x_1 < \dots, x_n$ and $y_1 < \dots, y_n$, then the permutation $\sigma$ is often either an identity mapping or an anti-identity mapping. This is illustrated in Figure 9.

It has been shown that one can construct counterexamples to this observation Beinert et al. (2022). However, in practice, the observation holds true in most relevant numerical settings Vayer et al. (2019b); Dumont et al. (2025). Specifically, it is shown in Dumont et al. (2025, Proposition 3.8) that this observation holds true under suitable conditions, providing evidence on why this obervation works well in practice. In these cases, the optimization problem in equation 11 reduces to ordering the support points. It can thus be solved at a computational cost of $\mathcal{O}(n \log(n))$.

### B.2    COUNTEREXAMPLES OF THE SORTING METHOD

The counterexample proposed in Beinert et al. (2022) is shown in the following proposition.

**Proposition B.1.** *Let the objective function be denoted as*

$$F_\sigma(x,y) := \frac{1}{n^2} \sum_{i,j} |x_i - x_j|^\alpha \cdot |y_{\sigma(i)} - y_{\sigma(j)}|^\alpha \tag{13}$$

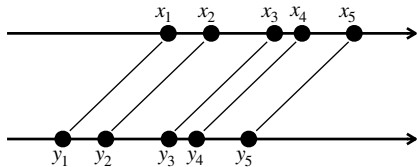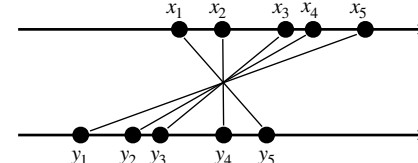

Figure 9: The optimal Gromov-Wasserstein matching in 1D is typically either an identity mapping or an anti-identity mapping.

*The identity mapping and anti-identity mapping result in the objective values denoted as $F_{id}(x, y)$ and $F_{a\text{-}id}(x, y)$.*

*For given $n > 3$ and $\alpha > 0$, the explicit instance is constructed by $x(\epsilon) = (x_i)_{i=1}^n$ and $y(\epsilon) = (y_i)_{i=1}^n$ with $\epsilon \in (0, 2/(n-3))$ given by*

$$
x_i := \begin{cases} -1, & i = 1, \\ \frac{2i-n-1}{2}\epsilon, & i = 2, \ldots, n-1, \\ 1, & i = n \end{cases} \quad \text{and} \quad y_i := \begin{cases} -1, & i = 1, \\ -1 + \epsilon, & i = 2, \\ (i-2)\epsilon, & i = 3, \ldots, n. \end{cases}
$$

*Such instance gives*

$$
F_{\text{id}}(x, y) < \max_{\sigma \in S_n} F_\sigma(x, y) \quad \text{and} \quad F_{\text{a}-\text{id}}(x, y) < \max_{\sigma \in S_n} F_\sigma(x, y).
$$

*Moreover, the gap can become arbitrary large for increasing $n \in \mathbb{N}$.*

### B.3    THE TRUE CASE WHEN $GW = 0$

Despite the counterexamples shown above, when the GW distance between two one-dimensional measures is **zero**, the optimal matching is always given by the identity or the anti-identity map, as shown by the following theorem.

**Theorem B.2.** *Let $x_1 < \cdots < x_n$ and $y_1 < \cdots < y_n$ be two strictly increasing sequences in $\mathbb{R}$, and let $\mu = \frac{1}{n}\sum_{i=1}^n \delta_{x_i}$ and $\nu = \frac{1}{n}\sum_{i=1}^n \delta_{y_i}$ associated with the Euclidean metric on $\mathbb{R}$. If $\text{GW}(\mu, \nu) = 0$, then the permutation $\sigma$ in equation 11 is the identity or anti-identitiy map.*

*Proof.* If $\text{GW}(\mu, \nu) = 0$, then the solution to the Gromov-Monge problem equation 11 is a permutation $\sigma$ such that $|x_i - x_j| = |y_{\sigma(i)} - y_{\sigma(j)}|$. This implies that for any $i < j < k$, $x_j$ lies between $x_i$ and $x_k$ are mapped to $y_{\sigma(j)}$ which also lies between $y_{\sigma(i)}$ and $y_{\sigma(k)}$. Repeat this for any consecutive triple $j - 1 < j + 1 < j + 2$, we obtain that $y_{\delta(j)}$ always lie between $y_{\sigma(j-1)}$ and $y_{\sigma(j+1)}$. Thus $\delta$ is monotone, either monotonic increasing or monotonic decreasing. That is, the permutation $\sigma$ is either the identity or the anti-identity map. □

## C    PROOFS

In this Section, we provide the proofs of our theoretical results.

### C.1    PROOF OF THEOREM 3.2

Note that both $GW = 0$ and $MSGW = 0$ hold trivially if the two measures are with $n \neq m$ or they are not endowed with uniform weights. Thus, it is sufficient to prove the theorem of two measures $\mu \in \mathbb{P}(\mathbb{R}^p)$ and $\nu \in \mathbb{P}(\mathbb{R}^q)$ with the same number of support points $n$ and with uniform weights, i.e.,

$$
\mu = \frac{1}{n}\sum_{i=1}^n \delta_{x_i}, \qquad \nu = \frac{1}{n}\sum_{i=1}^n \delta_{y_i}.
$$

### C.1.1 PROOF OF $MSGW = 0 \Longleftarrow GW = 0$

Since $GW(\mu, \nu) = 0$ and due to Definition 2.1, there exists a measure-preserving isometry between $\mu$ and $\nu$. Without loss of generality, we assume that $p < q$. The measure $\nu$ must then lie on a $p$-dimensional manifold in $\mathbb{R}^q$. Hence, the GW problem is equivalent to a GW problem between two measures on $\mathbb{R}^p$. Therefore, for this proof, it is sufficient to consider the case $p = q$. The measure-preserving isometry $f$ in Definition 2.1 is then a linear bijective map, which can be described by an orthogonal matrix $\Psi \in \mathbb{R}^{p \times p}$ (Berger, 2009, Theorem 9.1.3). More concretely, we adopt the following Lemma.

**Lemma C.1.** *An isometry of Euclidean space is an affine translation whose linear part is an orthogonal transformation.*

For a given projection direction $\theta \in \mathbb{S}^{p-1}$, we define $\phi = \Psi \theta \in \mathbb{S}^{p-1}$. Due to the isometric property of $\Psi$, for any $x \in \text{supp}(\mu)$, there is a $y \in \text{supp}(\nu)$ such that $y = \Psi x$. Also consider a second point $x' \in \text{supp}(\mu)$ and the corresponding $y' = \Psi x' \in (\mu)$. Then, it holds that

$$\left| \theta^T x - \theta^T x' \right| = \left| (\Psi \phi)^T x - (\Psi \phi)^T x' \right| = \left| \phi^T \Psi^{-1} x - \phi^T \Psi^{-1} x' \right| = \left| \phi^T y - \phi^T y' \right|,$$

where we used that $\Psi^T = \Psi^{-1}$. Thus, for all $\theta \in \mathbb{S}^{p-1}$ there is a $\phi \in \mathbb{S}^{p-1}$ such that $GW\left( (P_\theta)_\# \mu, (P_\phi)_\# \nu \right) = 0$.

Similarly, for any projection direction $\phi \in \mathbb{S}^{p-1}$, we can define $\theta = \Psi^{-1} \phi \in \mathbb{S}^{p-1}$, and with the points $x, x', y, y' \in \mathbb{R}^p$ defined as before, we get

$$\left| \phi^T y - \phi^T y' \right| = \left| \theta^T x - \theta^T x' \right|.$$

Thus, for all $\phi \in \mathbb{S}^{p-1}$ there is a $\theta \in \mathbb{S}^{p-1}$ such that $GW\left( (P_\theta)_\# \mu, (P_\phi)_\# \nu \right) = 0$.

It follows that $MSGW(\mu, \nu) = 0$.

### C.1.2 PROOF OF $MSGW = 0 \Longrightarrow GW = 0$

Moreover, in order to simplify notation, we assume without loss of generality that the measures are normalized to be zero-mean, i.e., $\sum_{i=1}^n x_i = 0$ and $\sum_{i=1}^n y_i = 0$. In the case of unnormalized measures, the results below hold with adding a shift $b \in \mathbb{R}^p$ in Lemma C.3 and the construction of the measure-preserving map.

To prove the result, we will use Theorem B.2 and the following two lemmas. More precisely, we study the maps

$$\Theta_\mu : \mathbb{S}^{n-1} \to \mathbb{P}(\mathbb{R}), \ \ \Theta_\mu(\theta) = (P_\theta)_\# \mu, \qquad \text{and} \qquad \Phi_\nu : \mathbb{S}^{n-1} \to \mathbb{P}(\mathbb{R}), \ \ \Phi_\nu(\phi) = (P_\phi)_\# \nu.$$

Note that the two sets $\boldsymbol{\mu}^{\mathbb{S}^{p-1}}$ and $\boldsymbol{\nu}^{\mathbb{S}^{q-1}}$ in equation 7 are the images of the maps $\Theta_\mu$ and $\Phi_\nu$, respectively. Moreover, note that the maps $\Theta_\mu$ and $\Phi_\nu$ are continuous with respect to the Wasserstein distance, since for small changes of $\theta$ (or, respectively, $\phi$) each support point of the 1D projection $\Theta_\mu(\theta)$ (or, respectively, $\Phi_\nu(\phi)$) moves smoothly in $\mathbb{R}$.

**Lemma C.2.** *The sets of elements in $\boldsymbol{\mu}^{\mathbb{S}^{p-1}}$ (defined in equation 7) that have less than $n$ support points are the images of the hyperspheres*

$$\mathbb{S}_{x,x'}^{p-2} = \{ \theta \in \mathbb{S}^{p-1} : \langle x - x', \theta \rangle = 0; \ x, x' \in supp(\mu) \}$$

*with respect to $\Theta_\mu$. Analogously, the sets of elements in $\boldsymbol{\nu}^{\mathbb{S}^{q-1}}$ (defined in equation 7) that have less than $n$ support points are the images of the $p - 2$ dimensional hyperspheres*

$$\mathbb{S}_{y,y'}^{p-2} = \{ \phi \in \mathbb{S}^{p-1} : \langle y - y', \phi \rangle = 0; \ y, y' \in supp(\nu) \}$$

*with respect to $\Phi_\nu$.*

*Proof.* Note that the projection $(P_\theta)_\# \mu$ has at most $n$ support points, and it has strictly less than $n$ support points if there are $x, x' \in \text{supp}(\mu)$ such that $\langle x - x', \theta \rangle = 0$, i.e., $x$ and $x'$ are projected onto the same point in $\mathbb{R}$. This gives rise to the hypersphere $\mathbb{S}_{x,x'}^{p-2}$. An analogous argument yields the expression for $\mathbb{S}_{y,y'}^{p-2}$. $\qquad \square$

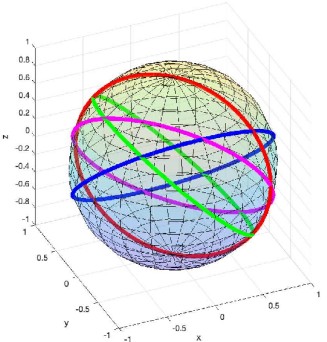

Figure 10: Sketch of the unit sphere $\mathbb{S}^2$ with four lower dimensional hypershperes $S^1_{x,x'}$.

It should be noted that a finite set of $p-2$ dimensional hyperspheres in $\mathbb{S}^{p-1}$ has measure zero with respect to the Lebesgue measure over $\mathbb{S}^{p-1}$, as illustrated for $p = 3$ in Figure 10. Thus, by the lemma, almost every element in $\boldsymbol{\mu}^{\mathbb{S}^{p-1}}$ and $\boldsymbol{\nu}^{\mathbb{S}^{q-1}}$ has $n$ support points, which leads us to the following result.

**Lemma C.3.** *Assume $MSGW(\mu, \nu) = 0$. Then, there exist linearly independent vectors $\theta_1, \ldots, \theta_p \in \mathbb{S}^{p-1}$ such that there is a permutation $\sigma : \{1, \ldots, n\} \to \{1, \ldots, n\}$ and linearly independent vectors $\phi_1, \ldots, \phi_p \in \mathbb{S}^{p-1}$ such that $\langle x_i, \theta_k \rangle = \langle y_{\sigma(i)}, \phi_k \rangle$, for all $i = 1, \ldots, n$ and $k = 1, \ldots, p$.*

*Proof.* Since $MSGW(\mu, \nu) = 0$, we know that for each $\theta \in \mathbb{S}^{p-1}$ there is a permutation $\sigma : \{1, \ldots, n\} \to \{1, \ldots, n\}$ and a vector $\phi \in \mathbb{S}^{p-1}$ such that $\langle x_i, \theta \rangle = \langle y_{\sigma(i)}, \phi \rangle$, for all $i = 1, \ldots, n$. We first show that there is a set of linearly independent vectors $\theta_1, \ldots, \theta_p \in \mathbb{S}^{p-1}$, for which the statement holds with *the same permutation $\sigma$*. Then we show that the corresponding vectors $\phi_1, \ldots, \phi_p \in \mathbb{S}^{p-1}$ are linearly independent.

1. *Existence of permutation:* Two support points $x, x' \in \mathbb{R}^p$ can change order in the 1D projection $\Theta_\mu(\theta)$, when $\theta$ crosses the hyperplane $\mathbb{S}^{p-2}_{x,x'}$ defined in Lemma C.2. However, by Theorem B.2, after crossing the hyperplane $\mathbb{S}^{p-2}_{x,x'}$, the optimal 1D GW map $\sigma$ must again be either the identity or the anti-identity map. Since each hyperplane $\mathbb{S}^{p-2}_{x,x'}$ has Lebesgue measure zero in $\mathbb{S}^{p-1}$ and there are only a finite number of hyperplanes, it follows that there exist linearly independent vectors $\theta_1, \ldots, \theta_p \in \mathbb{S}^{p-1}$ such that for some permutation $\sigma$ and some vectors $\phi_1, \ldots, \phi_p \in \mathbb{S}^{p-1}$ it holds $\langle x_i, \theta_k \rangle = \langle y_{\sigma(i)}, \phi_k \rangle$, for all $i = 1, \ldots, n$ and $k = 1, \ldots, p$.

2. *Linear independence of $\phi_1, \ldots, \phi_p$.* Define $Y_\sigma := \begin{bmatrix} y_{\sigma(1)} & \cdots & y_{\sigma(n)} \end{bmatrix} \in \mathbb{R}^{p \times n}$ and $u(\theta) := \begin{bmatrix} \langle x_1, \theta \rangle & \cdots & \langle x_n, \theta \rangle \end{bmatrix}^\top \in \mathbb{R}^n$. For each $k$, the identity $\langle x_i, \theta_k \rangle = \langle y_{\sigma(i)}, \phi_k \rangle$ for all $i$ is equivalent to the linear system

$$Y_\sigma^\top \phi_k = u(\theta_k). \tag{14}$$

By construction, $Y_\sigma$ has full row rank $p$, so equation 14 is consistent and its solution $\phi_k$ is unique. Now suppose $\sum_{k=1}^p a_k \phi_k = 0$. Multiply equation 14 by $a_k$ and sum over $k$: for every $i$,

$$0 = \Big\langle y_{\sigma(i)}, \sum_k a_k \phi_k \Big\rangle = \sum_k a_k \langle x_i, \theta_k \rangle = \Big\langle x_i, \sum_k a_k \theta_k \Big\rangle.$$

Since $\{x_i\}$ spans $\mathbb{R}^p$ and $\{\theta_k\}$ are linearly independent, this forces $a_k = 0$ for all $k$. Hence $\{\phi_k\}_{k=1}^p$ is linearly independent.

We are now ready to explicitly construct a measure-preserving map between $\mu$ and $\nu$. Based on the matrices $\Theta := \begin{bmatrix} \theta_1 & \cdots & \theta_p \end{bmatrix} \in \mathbb{R}^{p \times p}$ and $\Phi := \begin{bmatrix} \phi_1 & \cdots & \phi_p \end{bmatrix} \in \mathbb{R}^{p \times p}$ with the two bases defined by Lemma C.3, we consider the linear map $L = \Theta \Phi^{-1} \in \mathbb{R}^{p \times p}$. With this linear operator it holds $y_{\sigma(i)} = L^\top x_i$ for all $i = 1, \ldots, n$. Next, for each $\theta \in T$ choose a partner $\phi(\theta)$. For all $i$,

$$\langle x_i, \theta \rangle = \langle L^\top x_i, \phi(\theta) \rangle = \langle x_i, L\phi(\theta) \rangle.$$

Since $\{x_i\}$ spans $\mathbb{R}^p$, we get for all $\theta \in T$:

$$\theta = L\phi(\theta) .$$

Let $A := LL^\top \succ 0$ on $\mathbb{R}^p$. Then for all $\theta \in T$,

$$\langle A^{-1}\theta, \theta \rangle = \|L^{-1}\theta\|^2 = \|\phi(\theta)\|^2 = 1.$$

By the fact that a nonzero quadratic form cannot vanish on a positive-measure subset of the sphere, apply this to $B := A^{-1} - I$: if $A^{-1} \neq I$, then $\langle B\theta, \theta \rangle = 0$ could hold only on a set with measure zero. This contradicts to that the set $T$ has positive measure.

This forces $A^{-1} = I_p$, i.e.

$$LL^\top = I_p .$$

Thus, $L$ defines a measure-preserving map. $\qquad\square$

### C.2 Proof of Theorem 3.3

**Property 1.** (Positivity) $MSGW(\mu, \nu) \geq 0$.

The positivity follows directly from the positivity of the Gromov-Wasserstein distance.

**Property 2.** (Symmetry) $MSGW(\mu, \nu) = MSGW(\nu, \mu)$.

The symmetry follows from the symmetry of the two arguments in the max-operation in equation 4.

**Property 3.** (Triangle inequality) $MSGW(\mu, \nu) \leq MSGW(\mu, \gamma) + MSGW(\gamma, \nu)$.

It remains to show the triangle inequality. Given measures $\mu \in \mathbb{R}^p, \nu \in \mathbb{R}^q, \gamma \in \mathbb{R}^r$, by the triangle inequality for the Gromov-Wasserstein distance, it holds that

$$GW\left((P_\theta)_{\#}\mu, (P_\phi)_{\#}\nu\right) \leq GW\left((P_\theta)_{\#}\mu, (P_\delta)_{\#}\gamma\right) + GW\left((P_\delta)_{\#}\gamma, (P_\phi)_{\#}\nu\right), \qquad (15)$$

for all $\theta \in \mathbb{S}^{p-1}, \phi \in \mathbb{S}^{q-1}, \delta \in \mathbb{S}^{r-1}$. Taking the infimum over $\phi \in \mathbb{S}^{q-1}$ on both sides, it holds

$$\inf_{\phi \in \mathbb{S}^{q-1}} GW\left((P_\theta)_{\#}\mu, (P_\phi)_{\#}\nu\right) \leq GW\left((P_\theta)_{\#}\mu, (P_\delta)_{\#}\gamma\right) + \inf_{\phi \in \mathbb{S}^{q-1}} GW\left((P_\delta)_{\#}\gamma, (P_\phi)_{\#}\nu\right),$$

for all $\theta \in \mathbb{S}^{p-1}, \delta \in \mathbb{S}^{r-1}$. Then taking the infimum over $\delta \in \mathbb{S}^{r-1}$, we get

$$\inf_{\phi \in \mathbb{S}^{q-1}} GW\left((P_\theta)_{\#}\mu, (P_\phi)_{\#}\nu\right)$$

$$\leq \inf_{\delta \in \mathbb{S}^{r-1}} \left( GW\left((P_\theta)_{\#}\mu, (P_\delta)_{\#}\gamma\right) + \inf_{\phi \in \mathbb{S}^{q-1}} GW\left((P_\delta)_{\#}\gamma, (P_\phi)_{\#}\nu\right) \right)$$

$$\leq \inf_{\delta \in \mathbb{S}^{r-1}} GW\left((P_\theta)_{\#}\mu, (P_\delta)_{\#}\gamma\right) + \sup_{\delta \in \mathbb{S}^{r-1}} \inf_{\phi} GW\left((P_\delta)_{\#}\gamma, (P_\phi)_{\#}\nu\right),$$

for all $\theta \in \mathbb{S}^{p-1}$. Finally, taking the supremum over $\theta \in \mathbb{S}^{p-1}$ on both sides, yields

$$\sup_{\theta \in \mathbb{S}^{p-1}} \inf_{\phi \in \mathbb{S}^{q-1}} GW\left((P_\theta)_{\#}\mu, (P_\phi)_{\#}\nu\right) \leq \sup_{\theta \in \mathbb{S}^{p-1}} \inf_{\delta \in \mathbb{S}^{r-1}} GW\left((P_\theta)_{\#}\mu, (P_\delta)_{\#}\gamma\right)$$
$$+ \sup_{\delta \in \mathbb{S}^{r-1}} \inf_{\phi \in \mathbb{S}^{q-1}} GW\left((P_\delta)_{\#}\gamma, (P_\phi)_{\#}\nu\right). \qquad (16)$$

Similarly, taking the infimum over $\theta \in \mathbb{S}^{p-1}$ and $\delta \in \mathbb{S}^{r-1}$, and the supremum over $\phi \in \mathbb{S}^{q-1}$ in equation 15, we arrive at the bound

$$\sup_{\phi \in \mathbb{S}^{q-1}} \inf_{\theta \in \mathbb{S}^{p-1}} GW\left((P_\theta)_{\#}\mu, (P_\phi)_{\#}\nu\right) \leq \sup_{\phi \in \mathbb{S}^{q-1}} \inf_{\delta \in \mathbb{S}^{r-1}} GW\left((P_\delta)_{\#}\gamma, (P_\phi)_{\#}\nu\right)$$
$$+ \sup_{\delta \in \mathbb{S}^{r-1}} \inf_{\theta \in \mathbb{S}^{p-1}} GW\left((P_\theta)_{\#}\mu, (P_\delta)_{\#}\gamma\right). \qquad (17)$$

Taking the maximum of the two inequalities in equation 16 and equation 17 leads to

$$MSGW(\mu, \nu)$$

$$= \max \left( \sup_{\theta \in \mathbb{S}^{p-1}} \inf_{\phi \in \mathbb{S}^{q-1}} GW\left((P_\theta)_{\#}\mu, (P_\phi)_{\#}\nu\right), \sup_{\phi \in \mathbb{S}^{q-1}} \inf_{\theta \in \mathbb{S}^{p-1}} GW\left((P_\theta)_{\#}\mu, (P_\phi)_{\#}\nu\right) \right)$$

$$\leq \max \left( \sup_{\theta \in \mathbb{S}^{p-1}} \inf_{\delta \in \mathbb{S}^{r-1}} GW\left((P_\theta)_{\#}\mu, (P_\delta)_{\#}\gamma\right) + \sup_{\delta \in \mathbb{S}^{r-1}} \inf_{\phi \in \mathbb{S}^{q-1}} GW\left((P_\delta)_{\#}\gamma, (P_\phi)_{\#}\nu\right), \right.$$

$$\left. \sup_{\phi \in \mathbb{S}^{q-1}} \inf_{\delta \in \mathbb{S}^{r-1}} GW\left((P_\delta)_{\#}\gamma, (P_\phi)_{\#}\nu\right) + \sup_{\delta \in \mathbb{S}^{r-1}} \inf_{\theta \in \mathbb{S}^{p-1}} GW\left((P_\theta)_{\#}\mu, (P_\delta)_{\#}\gamma\right) \right)$$

$$\leq \max \left( \sup_{\theta \in \mathbb{S}^{p-1}} \inf_{\delta \in \mathbb{S}^{r-1}} GW\left((P_\theta)_{\#}\mu, (P_\delta)_{\#}\gamma\right), \sup_{\delta \in \mathbb{S}^{r-1}} \inf_{\theta \in \mathbb{S}^{p-1}} GW\left((P_\theta)_{\#}\mu, (P_\delta)_{\#}\gamma\right) \right)$$

$$+ \max \left( \sup_{\delta \in \mathbb{S}^{r-1}} \inf_{\phi \in \mathbb{S}^{q-1}} GW\left((P_\delta)_{\#}\gamma, (P_\phi)_{\#}\nu\right), \sup_{\phi \in \mathbb{S}^{q-1}} \inf_{\delta \in \mathbb{S}^{r-1}} GW\left((P_\delta)_{\#}\gamma, (P_\phi)_{\#}\nu\right) \right)$$

$$= MSGW(\mu, \gamma) + MSGW(\gamma, \nu)$$

### C.3 PROOF OF THEOREM 3.4

Note that the sets $\tilde{\mu} \in \boldsymbol{\mu}^{\mathbb{S}^{p-1}}$ and $\tilde{\nu} \in \boldsymbol{\nu}^{\mathbb{S}^{q-1}}$ are parameterized by the sets $\theta \in \mathbb{S}^{p-1}$ and $\phi \in \mathbb{S}^{q-1}$, respectively. More precisely, for each $\theta \in \mathbb{S}^{p-1}$ a one-dimensional measure $\tilde{\mu} \in \boldsymbol{\mu}^{\mathbb{S}^{p-1}}$ is constructed, and similarly for each $\phi \in \mathbb{S}^{q-1}$ a one-dimensional measure $\tilde{\nu} \in \boldsymbol{\nu}^{\mathbb{S}^{q-1}}$ is constructed. Thus, there are correspondences $(\theta, \tilde{\mu})$ and $(\phi, \tilde{\nu})$ such that $GW\left((P_\theta)_{\#}\mu, (P_\phi)_{\#}\nu\right) = GW\left(\tilde{\mu}, \tilde{\nu}\right)$. Therefore, we can rewrite the definition of MSGW in equation 4 as

$$MSGW(\mu, \nu)$$

$$= \max \left\{ \sup_{\theta \in \mathbb{S}^{p-1}} \inf_{\phi \in \mathbb{S}^{q-1}} GW\left((P_\theta)_{\#}\mu, (P_\phi)_{\#}\nu\right), \sup_{\phi \in \mathbb{S}^{q-1}} \inf_{\theta \in \mathbb{S}^{p-1}} GW\left((P_\theta)_{\#}\mu, (P_\phi)_{\#}\nu\right) \right\}$$

$$= \max \left\{ \sup_{\tilde{\mu} \in \boldsymbol{\mu}^{\mathbb{S}^{p-1}}} \inf_{\tilde{\nu} \in \boldsymbol{\nu}^{\mathbb{S}^{q-1}}} GW\left(\tilde{\mu}, \tilde{\nu}\right), \sup_{\tilde{\nu} \in \boldsymbol{\nu}^{\mathbb{S}^{q-1}}} \inf_{\tilde{\mu} \in \boldsymbol{\mu}^{\mathbb{S}^{p-1}}} GW\left(\tilde{\mu}, \tilde{\nu}\right) \right\}$$

$$= d_{\mathcal{H}}^{\mathbb{P}(\mathbb{R})} \left( \boldsymbol{\mu}^{\mathbb{S}^{p-1}}, \boldsymbol{\nu}^{\mathbb{S}^{q-1}} \right).$$

### C.4 PROOF OF PROPOSITION 4.2

Note that, similar to Theorem 3.4 the discrete approximation of MSGW is equivalent to a Hausdorff distance, $MSGW_{\Theta, \Phi}(\mu, \nu) = d_{\mathcal{H}}^{\mathbb{P}(\mathbb{R})}(\boldsymbol{\mu}^{\Theta}, \boldsymbol{\nu}^{\Phi})$. The result follows then from Mémoli (2011) (Remark 2.1), which states that for any continuous sets $A, B, \subset Z$ and their finite sample sets $\hat{A} \subset A$ and $\hat{B} \subset B$ it holds that

$$\left| d_{\mathcal{H}}^Z(A, B) - d_{\mathcal{H}}^Z\left(\hat{A}, \hat{B}\right) \right| \leq d_{\mathcal{H}}^Z\left(A, \hat{A}\right) + d_{\mathcal{H}}^Z\left(B, \hat{B}\right).$$

## D ADDITIONAL EXPERIMENTAL RESULTS

### D.1 ADDITIONAL RESULTS FOR REFLECTION AND ROTATION INVARIANCE

This section shows additional results of using GW, SGW, RISGW, and MSGW with spiral datasets, as a complement to Figure 2.

Figure 11 shows that MSGW is more stable with respect to discrete sampling compared with SGW. In particular, with an increasing number of projections (from $L = 10$ to $L = 30$), though both MSGW and SGW gain more stability, MSGW shows a much greater gain.

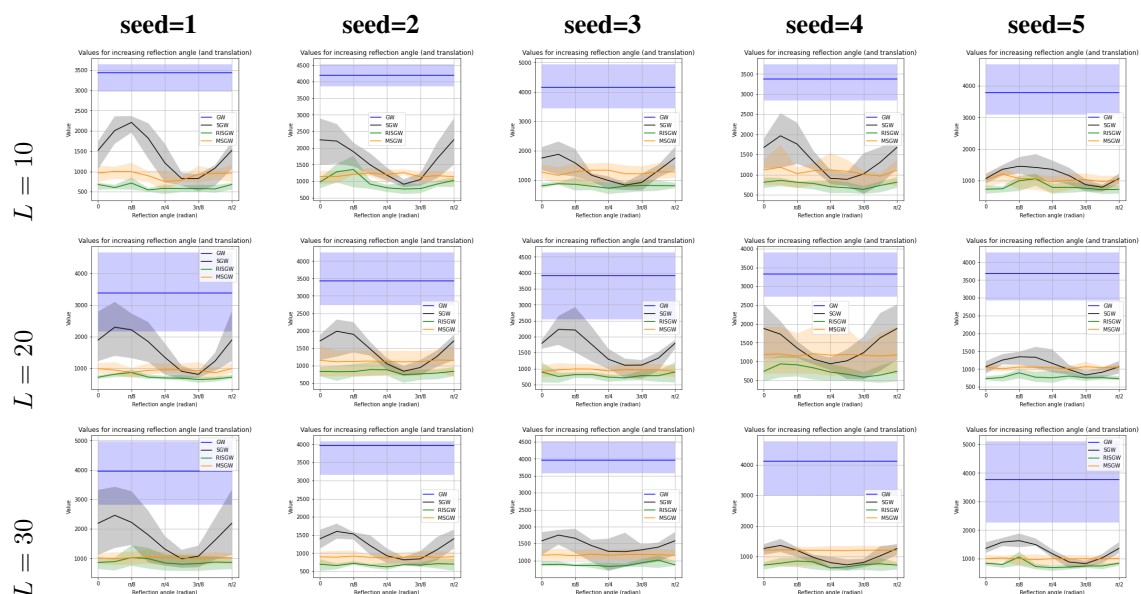

Figure 11: **Reflection invariance:** Comparison across different number of projection directions $L$ and different random seeds.

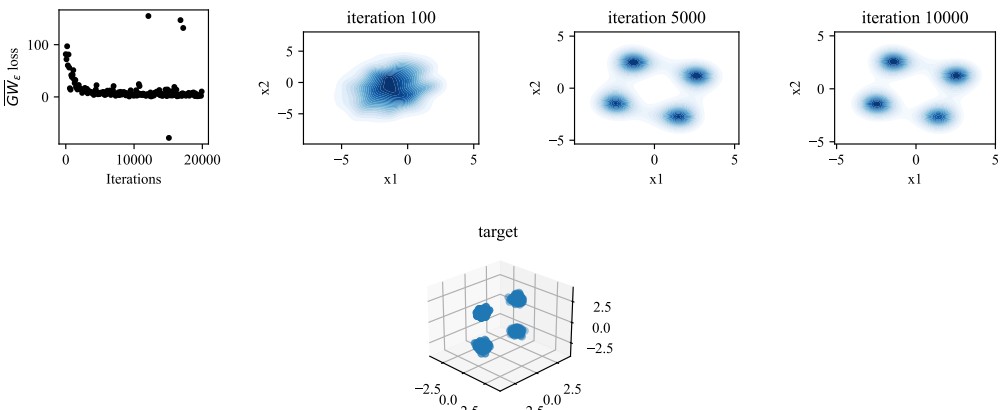

Figure 12: **GAN with GW loss.** $L = 20$ and 3D-4mode Gaussian target.

## D.2 ADDITIONAL RESULTS FOR GAN

This section shows additional results of using GW, SGW, and MSGW in GAN, as a complement to Figure 5.

**3D-4mode for GW and SGW**    We repeat the same experiments on GANs as introduced in Section 5 for GW and SGW with $L = 20$. Results are shown in Figure 12 and 13 for the GAN with GW loss and SGW loss, respectively. As expected, the GAN with GW loss converges quickly with respect to the number of iterations and generates well-constructed datasets. However, for this specific experimental setting, the GAN with SGW loss fails to generate high-quality datasets even with 20,000 iterations.

**Larger number of projections for SGW**    We tried to remedy the poor performance of the GAN with SGW loss by using a larger number of projection directions $L$. Figure 14 shows the experimental results when using $L = 20, 400, 1000$ number of projections. Although the evolution of the

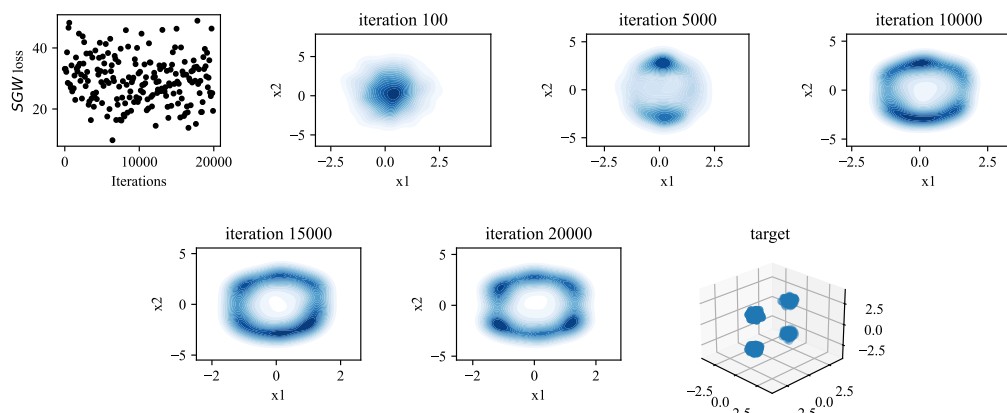

Figure 13: **GAN with SGW loss.** $L = 20$ and 3D-4mode Gaussian target.

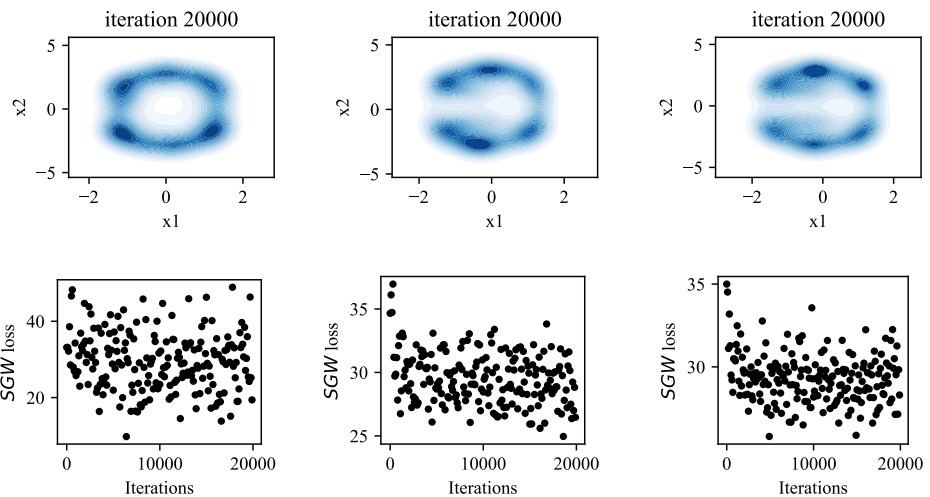

Figure 14: **GAN with SGW loss** with different number of projections (columns from left to right) $L = 20, 400, 1000$, and 3D-4mode Gaussian target.

loss values seems to converge more quickly as the number of projections increases, limited to the current experiment settings, it still fails to generate satisfying datasets.

**Generate 2D data from 4D Gaussian**   Figures 15 and 16 show the performance of using a SGW and MSGW loss in a GAN, where the target dataset is a 4D-4mode Gaussian. Notw that we are thus not able to plot the 4D data points in the figures. The number of projections is also set as $L = 20$. We can see that the GAN with MSGW loss still performs well in this higher-dimensional case.

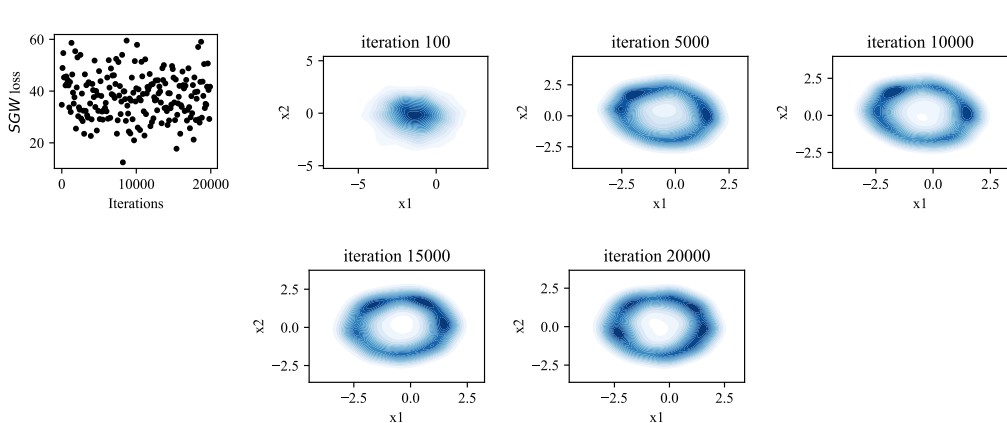

Figure 15: **GAN with SGW loss** with $L = 20$ and 4D-4mode Gaussian target.

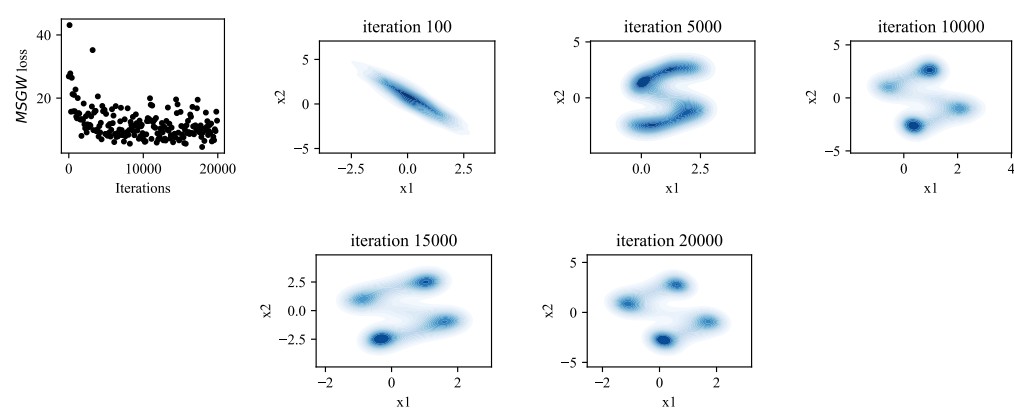

Figure 16: **GAN with MSGW loss** with $L = 20$ and 4D-4mode Gaussian target.

