# OpenReview forum: "Max-Min Sliced Gromov-Wasserstein"
_ICLR.cc/2026/Conference — Submitted to ICLR 2026_

### Official Review · Reviewer_LK3E · 2025-10-29

**Soundness:** 2
**Presentation:** 3
**Contribution:** 2
**Rating:** 2
**Confidence:** 5

**Summary:**

The paper presents a max–min formulation of a sliced Gromov–Wasserstein (SGW) variant that aims to avoid optimization on the Stiefel manifold while (theoretically) preserving rotational-invariance properties, and more importantly isometric invariance properties. The idea is to use a max-min formulation and to optimize only over directions in the hypersphere.

**Strengths:**

The manuscript contains several interesting contributions: the metric property stated in Theorem 3.4 is notable, since prior work only established a one-directional result for the basic SGW; and the connection to the Hausdorff distance between projected measures (Theorem 3.6) is also interesting. Moreover, the definition of MSGW itself is sound and quite elegant, as it theoretically preserves all the desirable properties of the original Gromov–Wasserstein (GW) formulation while avoiding optimization on the Stiefel manifold.
The literature review is thorough, and the paper is generally well written.

**Weaknesses:**

Despite these strengths, the paper based itself on a false premise. The authors rely on a closed-form expression for one-dimensional Gromov problems; however, recent work (acknowledged in [1]) shows that this closed form is incorrect — there is no general sorting algorithm that solves the 1-D Gromov problem. What the manuscript actually computes is therefore not the true GW distance but a related divergence that essentially evaluates the GW loss on either the identity or anti-identity permutation. Appendix B tells that these two permutations commonly appear in practice, but this empirical claim is not supported by a theoretical statement. The paper should explicitly acknowledge at the beginning that the computed quantity is a different divergence, and clarify its interest (see below).

Another concern relates to the novelty of the paper. Although the proposed formulation is interesting, it remains overall rather incremental compared to [1]. This issue is most apparent in the experimental section, which exactly reproduces the same setups and results as in [1], providing very little additional insight or validation.
I believe that including a comparison with the sliced Wasserstein distance could significantly strengthen the experimental part. One could define an analogous metric by replacing GW with W in the formulation, which would help clarify the true added value of this approach. Positioning the proposed method relative to sliced Wasserstein would better highlight the practical and conceptual interest of building a “sliced” metric based on an approximation of GW (using the identity or anti-identity permutation).

Finally a last concern is algorithmic and numerical: MSGW is formulated as a max–min problem, unlike RISGW. Max–min problems are notoriously difficult to optimize in practice, and the paper does not discuss optimization stability or convergence. Convergence curves, sensitivity to initialization, and practical runtime/iteration counts would be important to assess optimization behavior and the claimed benefits of avoiding the Stiefel manifold.

[1] Sliced Gromov-Wasserstein, Titouan Vayer, Rémi Flamary, Romain Tavenard, Laetitia Chapel, Nicolas Courty, arxiv 2022.

**Questions:**

Appart from the previous remarks I have technical concerns about the argument in Appendix C.2.1 that GW = 0 implies MSGW = 0. The proof appeals to the claim that a measure-preserving isometry is linear and bijective and thus represented by an orthogonal matrix. I believe that this seems to implicitly invoke results like the Mazur–Ulam theorem, which require hypotheses: in particular, one typically needs Euclidean metrics on both spaces for GW, and Mazur–Ulam gives that surjective isometries between normed vector spaces are affine not linear, so one must ensure the map sends 0 to 0 to conclude linearity.
More importantly, the argument should check issues about supports of the measures: I believe Mazur–Ulam works between spaces that are vector spaces. So in this case, in order to work, I believe the support of the measures should be the entire space $\mathbb{R}^p$ (which is not the case for discrete measures). I suggest the authors (i) make the exact assumptions explicit, (ii) cite the appropriate functional-analytic result because I could not find (Berger, 2009, Theorem 9.1.3) (I belive it is like Mazur–Ulam) and state whether they require the map to be affine or linear, and (iii) discuss any needed assumptions on supports of the measures.

Minor remark: stating “GW is $O(n^3)$” is misleading. The GW problem is NP-hard in general; the cubic runtime refers to a particular algorithmic routine (e.g., a cubic implementation of the GW solver per iteration) rather than complexity of solving GW exactly.

---

> ### Author Response · Authors · 2025-11-25
> **Response to Reviewer LK3E (part 1)**
>
> ***On solving 1D GW exactly***\
> We apologize for any potentially unrigorous statements regarding the use of the sorting method to solve the 1D GW.
> As we discuss in Appendix B, the solution to the 1D GW distance is indeed not always an identity or anti-identity, and therefore we write in Section 4 that this distance can "typically" be found by a sorting algorithm. We agree with the reviewer that this does not address the limitations clearly enough. Thus, in the revised paper, we will clarify the limitations already in the introduction and also in Section 4.
>
> However, we want to emphasize that our main contributions are theoretical and consist of the definition of MSGW and its metric properties.
> Note that all theoretical results (metric properties, equivalence with GW, Hausdorff interpretation, and the finite-direction error bound in Proposition 4.2) hold regardless of how the global solution to the 1D GW distance is found.
> While currently all sliced GW variants rely on the sorting algorithm, they could all be combined with methods for finding the globally optimal solution to the 1D GW problem (e.g., the low-dimensional global methods of Ryner et al., 2023).
>
> Finally, we want to note that the counterexamples provided in (Beinert et al., 2022;
> Dumont et al., 2025) are highly idealistic and are hardly observed in practice.
> Dumont et al. also discovered that under certain structural assumptions on the measures, monotone maps are optimal in 1D. They numerically observe that monotone maps are often optimal in non-pathological settings.
> Thus, we do not consider these counterexamples to undermine the practical effectiveness of this sorting method, at least so far.
> Theorem B.1. and Figure 8 show that the result does indeed hold when the GW distance is 0.
> To give a more complete picture in the revised version of the manuscript, we will also add a short summary of the counterexamples in the two papers (Beinert et al., 2022;
> Dumont et al., 2025) in Appendix B.
>
> ***About the overall novelty***\
> We argue that MSGW is a significant improvement on SGW, even though our work is largely based on the previous work on SGW.
> More precisely, we propose a max–min game that allows different projection directions for the two measures, which
> 1) recovers full isometric invariance in a sliced framework，
> 2) avoids any optimization on the Stiefel manifold，
> 3) leads to a Hausdorff-type interpretation as a pseudo-Hausdorff distance between sets of 1D projections (Theorem 3.6).
>
> Under standard discrete assumptions, we prove MSGW is a metric up to measure-preserving isometries, i.e., it induces exactly the same equivalence relation as GW on mm-spaces. Prior work on SGW only showed a one-directional connection and lacked the equivalence of zero sets.
>
> We derive an error bound for approximating MSGW with finite projections, via a Hausdorff-type argument over direction sets, and we evaluate this bound numerically (Figure 6). This provides both theoretical and empirical insight into how MSGW converges as the number of directions grows, which is not present in the SGW paper.
>
> On the experimental side, we indeed use similar toy setups to SGW (spirals, horses) intentionally, to make comparisons meaningful. But we also add some new experiments, in particular testing the MSGW in higher dimensions and with different number of projections.
>
> ***About replacing GW with W in the same formulation***\
> Note that a max-min sliced Wasserstein distance would have some rotation and reflection invariance properties, but \emph{it would not be translation invariant}.
> While we think the suggestion of replacing GW with W can be meaningful in some ways and is an interesting research direction, we consider it unnecessary and out of scope for our current work, where we aim to improve on the SGW distance (which is translation invariant).
>
> ***About the max-min optimization***\
> We acknowledge that the questions of max-min optimization in a ***continuous*** space represent meaningful research directions. In this work we are taking a direct sampling approach that approximates the MSGW with ***discrete*** sampling. We observed from the experiments that with very light computation with around $L=20$, the results are satisfactory enough. In some real-world scenarios, it may be necessary to explicitly develop a max-min optimization algorithm, but we leave this to future work.

---

> ### Author Response · Authors · 2025-11-25
> **Response to Reviewer LK3E (part 2)**
>
> ***About the proof of GW = 0 implies MSGW = 0***\
> We admit that the Mazur–Ulam theorem is common, but as you mentioned, it is not suitable for this discrete case. Instead, we use another theorem (in Berger 2009) explicitly for this discrete case, which allows us to directly use the orthogonal matrix. We will quote the following result in Appendix C.2.1.
>
>
> Proposition 9.1.3. in Berger, 2009: A map $f: X \rightarrow X$ lies in Is $(X)$ if and only if $f \in \mathrm{GA}(\dot{X})$ and $\vec{f} \in O(\vec{X})$.
>
> In our case, since $X$ is a Euclidean space, $\mathrm{GA}(\dot{X})$ is the set of affine maps, $x \to Ax + b$, where $A$ is invertible. For the map to be in $O(\vec{X})$, $A$ must also be an orthogonal matrix. Since we have normalized the measures to have mean zero, we can assume without loss of generality that $b=0$.
> Note that this is possible due to the translation invariance of the GW distance.
> Thus, any isometric map can be directly associated with an orthogonal matrix.
>
> ***About the complexity of GW***\
> Sorry for the confusion, this statement is indeed not precise enough. The general GW is NP-hard if we are aiming for the global optimal solution. The naive method for finding local solutions in discrete GW is $\mathcal{O}(n^4)$.
>
> The complexity $\mathcal{O}(n^3)$ usually refers to a widely-used method for discrete GW with the family of decomposable functions. It has been widely used through POT (Python Optimal Transport) package. The original paper proposed the algorithm (please see Remark 1 for the complexity analysis) is "Gabriel Peyre, Marco Cuturi, and Justin Solomon. Gromov-Wasserstein averaging of kernel and distance matrices. In Proceedings of The 33rd International Conference on Machine Learning,
> pp. 2664–2672. PMLR, 2016.".
> The same paper also states that the complexity for GW is $\mathcal{O}(n^4)$ using the naive implementation.
>
> In the same paper, with the projected gradient descent algorithm, the complexity of entropic GW is also the order of $\mathcal{O}(n^3)$. Although within each optimization iteration, Sinkhorn is used with $\mathcal{O}(n^2)$, the gradient is computed (still for the family of decomposable functions) with $\mathcal{O}(n^3)$. Thus the overall complexity is $\mathcal{O}(n^3)$.

---

### Official Review · Reviewer_d8H1 · 2025-10-30

**Soundness:** 3
**Presentation:** 4
**Contribution:** 4
**Rating:** 8
**Confidence:** 4

**Summary:**

This paper introduces the max-min Sliced Gromov-Wasserstein (MSGW) discrepancy as a computationally efficient surrogate for the Gromov-Wasserstein (GW) distance. It overcomes limitations of previous sliced GW methods, which are either computationally expensive or sacrifice rotational and reflection invariance.

**Strengths:**

- The novelty is simple yet clever, and well explained (even with the aid of a figure): there is no need to embed the measures into a common space (as for RISGW), since different projections are used. Thus, the idea aligns more naturally with the fundamentals of GW and is well-suited for future generalizations beyond measures supported on Euclidean spaces.

- Theoretical results are clearly presented. They study important properties of the methodology, including the metric property, the relation with the Hausdorff distance, and the error incurred by finite sample approximations.

- Several well-designed experiments demonstrate the rotation and reflection invariance of MSGW, its efficiency compared with existing GW variants, and its performance as a loss function in the generator of a GAN architecture. In addition, sensitivity analyses under different numbers of slices and noisy data are provided, showing both error control (utilizing theoretical error bounds) and robustness.

**Weaknesses:**

(a) Assumptions 3.3 for Theorem 3.4 are rather strong:

1 - The measures must be essentially supported on the same ambient space. This undermines the fundamentals of GW, where the ambient space should not matter. Moreover, under this assumption no embedding $\Delta$ is required.

2 - The measures must be empirical with the same number of points.

(b) The main text would benefit from including short sketches of the proofs of the main results (theorems and propositions).

**Questions:**

- Table 1, row 1: How is the $\mathcal O(n^3)$ complexity calculated? Solvers for GW usually achieve $\mathcal O(n^4)$; see, for example, Kerdoncuff et al., Sampled Gromov-Wasserstein. I would also suggest adding the regularized version of GW (entropic GW) to such table, since it is later used in the experiments section.
- The cited paper by Vayer et al. on SGW contains an error acknowledged by those authors in a revised version. Is this what the authors of this manuscript intended to refer to in line 171?
- What can the authors say about translation invariance?
- Can the authors comment on a possible dynamical framework as a byproduct of their work, if any?
- Why is RISGW not included in Figures 6 and 7?
- Is there any theoretical evidence connecting MSGW and GW, such as equivalence results or bounds?

Typos and stylistic issues:
- Revise quotation mark symbols.
- Footnote 3 lacks a period.
- Revise punctuation in line 353.

**Details Of Ethics Concerns:**

No concers.

---

> ### Author Response · Authors · 2025-11-25
> **Response to Reviewer d8H1**
>
> ***On Assumption 3.3*** \
> We apologize for any confusion that arises from **Assumption 3.3, Theorem 3.4, and Remark 3.5**.
>
> First of all, we note that the reverse implication of Theorem 3.4 does not require Assumption 3.3 to hold (cf. last sentence of Remark 3.5).
> To clarify this, we have therefore decided to split the two implications in Theorem 3.4 into two separate results.
>
> Moreover, the assumptions are chosen to simplify the proof, but they are indeed overly conservative and can be greatly relaxed:
> 1) As discussed in remark 3.5, we do not need to require that $p=q$.
> 2) If the measures $\mu$ and $\nu$ do not have the same number of support points, then Theorem 3.4 holds trivially, since neither $MSGW(\mu, \nu)=0$, nor $GW(\mu, \nu)=0$.
> 3) The assumption of uniform weights over the support points simplifies the proof substantially; we therefore prefer to keep it, but stress that this is purely assumed for simplified notation.
>
> We will revise this part of the paper to remove unnecessary assumptions and clarify where the assumptions are introduced only for technical reasons.
>
> Finally, we note that theorem 3.4 identifies the equivalence classes of MSGW, but this result does not mean that MSGW is only meaningful in this regime. Outside the assumption, MSGW remains a pseudo-metric and can still be used exactly as GW is in practice. In other words, dropping the assumption, only the equivalence proof is lost, not the definition.
>
> ***Other questions***
> 1. *About the complexity of GW*
>
>     Sorry for the confusion, this statement is indeed not precise enough. The general GW is NP-hard if we are aiming for the global optimal solution. The naive method for finding local solutions in discrete GW is $\mathcal{O}(n^4)$, as you pointed out.
>
>     The complexity $\mathcal{O}(n^3)$ usually refers to a widely-used method for discrete GW with the family of decomposable functions. It has been widely used through POT (Python Optimal Transport) package. The original paper proposed the algorithm (please see Remark 1 for the complexity analysis) is "Gabriel Peyre, Marco Cuturi, and Justin Solomon. Gromov-Wasserstein averaging of kernel and distance matrices. In Proceedings of The 33rd International Conference on Machine Learning, pp. 2664–2672. PMLR, 2016.".  The same paper also states that the complexity for GW is $\mathcal{O}(n^4)$ using the naive implementation.
>
>     In the same paper, with the projected gradient descent algorithm, the complexity of entropic GW is also the order of $\mathcal{O}(n^3)$. Although within each optimization iteration, Sinkhorn is used with $\mathcal{O}(n^2)$, the gradient is computed (still for the family of decomposable functions) with $\mathcal{O}(n^3)$. Thus the overall complexity is $\mathcal{O}(n^3)$.
>
>     We will include the paper you mentioned as a reference in the introduction of our manuscript and add the complexity of entropic GW to the table.
>
> 2. *The cited paper by Vayer et al. on SGW contains an error acknowledged by those authors in a revised version. Is this what the authors of this manuscript intended to refer to in line 171?*\
> Yes, exactly.
>
> 3. *What can the authors say about translation invariance?*\
> Thank you for pointing it out. We indeed need to clarify this simple yet crucial point: MSGW is translation-invariant - just as all other GW variants. This is, in fact, also illustrated in the experiment shown in Figure 2. The target datasets are created with both reflection and translation, though in the previous manuscript, we only titled it as "Reflection invariant". We will modify the text to specifically point out the translation invariance as well.
>
>
> 4. *Can the authors comment on a possible dynamical framework as a byproduct of their work, if any?*\
> Could you please clarify what you mean by a dynamical framework?
> We are aware of "dynamical optimal transport" as opposed to "static optimal transport", but this seems like an orthogonal direction to this paper.
>
> 5. *Why is RISGW not included in Figures 6 and 7?*\
> Good point. We will add RISGW in Figure 6 and 7.
>
> 6. *Is there any theoretical evidence connecting MSGW and GW, such as equivalence results or bounds?*\
> We agree that this is a very meaningful direction, but we leave it to future work.
>
> Additionally, we have corrected the typos as pointed out.

---

### Official Review · Reviewer_pkqq · 2025-10-31

**Soundness:** 3
**Presentation:** 3
**Contribution:** 3
**Rating:** 4
**Confidence:** 3

**Summary:**

Problem: Comparing data that live in different metric spaces is expensive with the Gromov–Wasserstein (GW) distance (non-convex). Existing sliced approximations speed things up but either lose rotation/reflection invariance (SGW) or become computationally costly/unstable when they try to restore it.

Motivation: Keep the speed of slicing while preserving the isometric properties of the original GW, and avoid the high-dimensional Stiefel-manifold optimization used by RISGW.

Contributions: (1) Propose Max–min slicing game: For each projection of one distribution, choose the “best” projection of the other to minimize 1D GW; then take the worst case over directions, and symmetrize. This yields a rotation/reflection-invariant sliced discrepancy. (2) Under standard discrete, uniform conditions, MSGW is a metric up to measure-preserving isometries (same as GW). In general it is a pseudo-metric. (3) Computation: Use a finite set of directions and evaluate all pairs (LxL) of 1D GW problems. Complexity is O(L^2 n log n) (slightly above SGW’s O(L n log n), far below RISGW). An error bound quantifies the approximation from finite projections.

Experiments are simple, not much practical.

**Strengths:**

- It aims for SGW-like cost while avoiding the expensive Stiefel-manifold optimization of RISGW; experiments show modest runtime vs RISGW and that entropic GW can hit memory limits.
- The adversarial slicing game is conceptually simple: minimize 1D GW for paired projections and then take the worst case; this is what yields the invariance while staying efficient.
- The paper proves MSGW preserves GW’s metric properties (up to measure-preserving isometries) and gives a finite-projection error bound; it also interprets MSGW as a pseudo Hausdorff distance between sets of 1D projections.
- To the authors’ knowledge, this is the first sliced GW that keeps rotation/reflection invariance while remaining computationally efficient.

**Weaknesses:**

- Assumption-heavy theory. Metric guarantees (“MSGW is a metric up to measure-preserving isometries”) rely on Assumption 3.3 and specific conditions; outside these, the distance is only a pseudo-metric.
- The experimental setting is poor. I don't understand why the authors try to test the method on GAN experiment. What are the reasons/motivations of experimenting the new method on GAN?
- Limited empirical scope. Experiments focus on spiral point clouds, a horse-mesh dataset, and a GAN toy setup; there’s no large-scale real-world benchmark to test scalability or downstream tasks.

**Questions:**

- What is the motivation for evaluating MSGW specifically in a GAN setup? Which property of MSGW (e.g., rotation/reflection invariance, noise robustness) is the GAN test meant to stress?

- Please provide quantitative GAN results for the MSGW-loss experiment, not just loss value. Report across several L values and random seeds.

- Can you add a large-scale, real-world benchmark (e.g., point-cloud registration on ModelNet/ShapeNet) to test scalability and downstream accuracy?

- The experiments are somewhat limited. Can you think of other useful downstream tasks and provide it?

---

> ### Author Response · Authors · 2025-11-25
> **Response to Reviewer pkqq**
>
> ***On Assumption 3.3***
>
> We apologize for any confusion that arises from **Assumption 3.3, Theorem 3.4, and Remark 3.5**.
>
> First of all, we note that the reverse implication of Theorem 3.4 does not require Assumption 3.3 to hold (cf. last sentence of Remark 3.5). To clarify this, we have therefore decided to split the two implications in Theorem 3.4 into two separate results.
>
> Moreover, the assumptions are chosen to simplify the proof, but they are indeed overly conservative and can be greatly relaxed:
>
> 1. As discussed in Remark 3.5, we do not need to require that \(p = q\).
> 2. If the measures $\mu$ and $\nu$ do not have the same number of support points, then Theorem 3.4 holds trivially, since neither $MSGW(\mu, \nu) = 0$ nor $GW(\mu, \nu) = 0$.
> 3. The assumption of uniform weights over the support points simplifies the proof substantially; we therefore prefer to keep it, but emphasize that this is purely for simplified notation.
>
> We will revise this part of the paper to remove unnecessary assumptions and clarify where they are introduced only for technical reasons.
>
> Finally, we note that Theorem 3.4 identifies the equivalence classes of MSGW, but this result does not mean that MSGW is only meaningful in this regime. Outside the assumption, MSGW remains a pseudo-metric and can still be used exactly as GW is in practice. In other words, dropping the assumption only loses the equivalence proof, not the definition.
>
> ***About GAN experiment***
>
> Our motivation for this experiment is to show that MSGW is not only a theoretically principled distance but also a practical loss for generative modeling across incomparable spaces, similar in spirit to GW-based generative models. Prior work has already used GW as the core loss in generative models across different domains and dimensions. Like GW, MSGW compares distributions via pairwise relational structure and does not require a common ambient space. In our toy setup, we deliberately construct target distributions that differ by rigid transformations; a non-invariant sliced distance (SGW) would penalize these, while MSGW respects them. This is particularly important in applications such as shape generative models, where the same object can appear in multiple poses. The max–min design of MSGW reduces the variance coming from random directions, compared to SGW. The GAN experiment shows that MSGW can produce stable training curves and visually plausible samples when used as a generator loss (while SGW fails), without incurring the heavy cost of full GW.
>
> ***About conducting other large-scale experiments***
>
> We have focused on the experiments to showcase the different theoretical properties that MSGW possesses, and also compare to those in the SGW paper. Due to the limited time available during the discussion phase and the strong focus on the theoretical contributions of this work, we have to leave the large-scale experiments and experiments for other downstream tasks for future work.

---

### Official Review · Reviewer_hprL · 2025-11-01

**Soundness:** 3
**Presentation:** 3
**Contribution:** 2
**Rating:** 4
**Confidence:** 4

**Summary:**

**Summary**
This paper proposes the **Max–Min Sliced Gromov–Wasserstein (MSGW)** distance, a new sliced approximation to the GW distance that is both **rotation/reflection invariant** and **computationally efficient**. The method introduces a **max–min formulation** over two different projection directions.

By allowing different projection directions for the two measures, MSGW restores the isometric invariance lost in standard Sliced GW (SGW) while avoiding the high cost of Rotation-Invariant SGW (RISGW). The overall complexity is $O(L^2 n \log n)$, and the distance can be interpreted as a Hausdorff metric between sets of 1D GW projections. Experiments show that MSGW achieves rotation-invariant, stable performance and compares favorably with SGW and RISGW in both accuracy and efficiency.

**Strengths:**

Strengths
The paper introduces a novel and well-justified formulation, Max–Min Sliced Gromov–Wasserstein (MSGW), which effectively resolves the long-standing limitation of rotation and reflection sensitivity in sliced GW methods. The proposed max–min structure allows the two measures to project onto different directions, achieving isometric invariance while maintaining linear-time complexity in each 1D GW computation. This formulation is theoretically elegant, connecting MSGW to the Hausdorff distance over sets of one-dimensional GW projections. The paper provides solid analytical results, including invariance proofs and conditions under which MSGW coincides with GW, offering a clear theoretical foundation.

Empirically, the experiments demonstrate that MSGW achieves the intended invariance and stability without the heavy optimization cost of RISGW. On both synthetic and geometric datasets, MSGW consistently outperforms SGW in robustness to rotations, reflections, and noise, while remaining computationally efficient. The results confirm that MSGW is a practical and theoretically principled alternative to existing sliced GW variants, with strong potential for downstream applications in shape matching and generative modeling.

**Weaknesses:**

Theoretical Weakness
In Remark 4.1 and Appendix B, the authors claim that the 1D GW problem can be solved in $O(n \log n)$ time by sorting or anti-sorting the projected points. However, this statement is not theoretically valid in general. The assumption that sorting provides the exact optimizer holds only in very specific cases and lacks general proof. In fact, it has been shown that 1D GW does not admit a closed-form solution based solely on sorting. Counterexamples have been rigorously presented in “On Assignment Problems Related to Gromov–Wasserstein Distances on the Real Line”, demonstrating that the optimal permutation can differ from both the identity and reverse mappings. Moreover, even the original Sliced Gromov–Wasserstein paper (Vayer et al., 2019) — cited in this work — later acknowledged that its main theorem supporting this property was disproved.

Therefore, the reliance on sorting or anti-sorting as an “exact” solver for 1D GW introduces a significant theoretical inconsistency. While it may work as a practical heuristic, presenting it as an exact and universally efficient solution undermines the rigor of the proposed complexity analysis and the claimed $O(n \log n)$ efficiency. The paper would be stronger if it explicitly discussed this limitation and clarified whether MSGW’s theoretical properties still hold when the 1D subproblems are only approximately solved.

- Lack of approximation error analysis.
The paper introduces a finite-direction approximation of MSGW but does not provide a clear analysis of how the approximation behaves as the number of sampled directions $L$ varies. In theory, MSGW replaces continuous optimization over the unit sphere with discrete sets of directions $(\Theta, \Phi)$, but the paper does not quantify the rate of convergence or the sensitivity of MSGW to the number of sampled directions. This is particularly relevant in high-dimensional settings, where the number of directions required for a stable estimate can grow rapidly, potentially offsetting the claimed computational benefits.

Empirically, no ablation studies are provided to show how the choice of $L$ affects accuracy, invariance quality, or runtime. Without such analysis, it is difficult to assess the robustness of MSGW under realistic computational constraints. A more systematic investigation of how direction sampling impacts performance would make the method’s practical reliability much clearer.

**Questions:**

- How sensitive is MSGW to the number of sampled projection directions $L$ in practice? It would be useful to see ablation experiments showing how varying $L$ affects both computational cost and accuracy, especially in higher dimensions.

- The experiments mainly involve 2D synthetic and shape-matching tasks. Have the authors tested MSGW on more complex or higher-dimensional datasets (e.g., point clouds or graph domains) to evaluate its scalability and robustness in realistic scenarios?

Other comments:
Theorem 3.2 claims MSGW is pseudo-metric. In my opition, the author can claim it is a metric in a quotient space G/\sim, where  G  is the set of all mm-spaces with probability measure, \sim is the  equivlent relation defined by GW(X,Y)=0.
In Theorem 3.4 the authors prove MSGW=0 implies GW=0. Thus, MSGW can be treated as metric (where identity is defined by \sim ).

---

> ### Author Response · Authors · 2025-11-25
> **Response to Reviewer hprL**
>
> ***On solving 1D GW exactly***
>
> We apologize for any potentially unrigorous statements regarding the use of the sorting method to solve the 1D GW. As we discuss in Appendix B, the solution to the 1D GW distance is indeed not always an identity or anti-identity, and therefore we write in Section 4 that this distance can *“typically”* be found by a sorting algorithm. We agree with the reviewer that this does not address the limitations clearly enough. Thus, in the revised paper, we will clarify the limitations already in the introduction and also in Section 4.
>
> However, we want to emphasize that our main contributions are theoretical and consist of the definition of MSGW and its metric properties. Note that all theoretical results (metric properties, equivalence with GW, Hausdorff interpretation, and the finite-direction error bound in Proposition 4.2) hold regardless of how the global solution to the 1D GW distance is found. While currently all sliced GW variants rely on the sorting algorithm, they could all be combined with methods for finding the globally optimal solution to the 1D GW problem (e.g., the low-dimensional global methods of Ryner et al., 2023).
>
> Finally, we want to note that the counterexamples provided in Beinert et al. (2022) and Dumont et al. (2025) are highly idealistic and hardly observed in practice. Dumont et al. also discovered that under certain structural assumptions on the measures, monotone maps are optimal in 1D. They numerically observe that monotone maps are often optimal in non-pathological settings. Thus, we do not consider these counterexamples to undermine the practical effectiveness of this sorting method, at least so far.
> Theorem B.1 and Figure 8 show that the result does indeed hold when the GW distance is 0. To give a more complete picture in the revised version of the manuscript, we will also add a short summary of the counterexamples in the two papers (Beinert et al., 2022; Dumont et al., 2025) in Appendix B.
>
> ***On approximation error analysis with different projections (L)***
>
> Thank you for pointing this out. We are currently working on experiments with different choices of \(L\) in the spiral dataset and GAN experiments and will provide the results in the following days. In general, a very small value of \(L\) (around 20) will give us desirable results.
>
> ***About doing experiments in higher dimension***
>
> The reviewer said that “The experiments mainly involve 2D synthetic and shape-matching tasks.” However, we would like to note that we have also done experiments in 3D and 4D spaces. We will provide more experiments when we update the manuscript.
>
> ***On Theorems 3.2 and 3.4***
>
> The reviewer is correct. Theorem 3.2 and 3.4 together prove that MSGW is a metric in a quotient space as the reviewer describes (in our words, “up to measure preserving isometries”). We split the result into two theorems to make some simplifying assumptions (which however can be relaxed; see our comments on Assumption 3.3).
>
> ***On Assumption 3.3***
>
> We apologize for any confusion that arises from **Assumption 3.3, Theorem 3.4, and Remark 3.5**.
>
> First of all, we note that the reverse implication of Theorem 3.4 does not require Assumption 3.3 to hold (cf. last sentence of Remark 3.5). To clarify this, we have therefore decided to split the two implications in Theorem 3.4 into two separate results.
>
> Moreover, the assumptions are indeed overly conservative and can be greatly relaxed:
>
> 1. As discussed in Remark 3.5, we do not need to require that \(p = q\).
> 2. If the measures $\mu$ and $\nu$ do not have the same number of support points, then Theorem 3.4 holds trivially, since neither $MSGW(\mu, \nu) = 0$, nor $GW(\mu, \nu) = 0$.
> 3. The assumption of uniform weights over the support points simplifies the proof substantially; we therefore prefer to keep it, but emphasize that this is purely for simplified notation.
>
> We will revise this part of the paper to remove unnecessary assumptions and clarify which assumptions are introduced only for technical reasons.
>
> Finally, we note that Theorem 3.4 identifies the equivalence classes of MSGW, but this result does not mean that MSGW is only meaningful in this regime. Outside the assumption, MSGW remains a pseudo-metric and can still be used exactly as GW is in practice. In other words, dropping the assumption loses only the equivalence proof, not the definition.

---

### Author Response · Authors · 2025-11-25
**Sincere thank to the reviewers and the chairs for the feedback**

We thank the reviewers for the constructive and positive feedback on our work.
Below, please find responses to the individual points addressed by each reviewer. We are currently updating our experiments and proof to relax the assumptions in the manuscript. We will upload a new PDF within this week.

---

### Author Response · Authors · 2025-11-29
**New manuscript (version 2.0) uploaded**

We have uploaded a new manuscript (version 2.0). We thank all reviewers for their constructive feedback.

---

### Meta-Review · Area_Chair_B1ar · 2025-12-28

**Summary:**

The authors consider the sliced approach for Gromov-Wasserstein (GW) problem for measures supported in different metric spaces. In particular, the authors proposed the max-min sliced Gromov-Wasserstein (MSGW), which is not only fast for computation as in sliced-Gromov-Wasserstein (SGW), but also preserves the rotation and reflection invariant of GW.

The Reviewers think that the proposed MSGW is theoretical elegant. However, the Reviewers raise serious concerns on its based premise, i.e., the 1d-GW for its algorithmic approach. Although the authors provide interesting theoretical finding results for the MSGW, it is questionable whether such results are still hold for the discrete approximation with the heuristic on 1d-GW problem. Overall, we think that the submission is below the bar. The authors may rely on several comments of the Reviewers to improve the submission.

**Reviewer Concerns:**

The Reviewers have some following concerns:

+ Reviewer hprL: theoretical weakness on the computational complexity claims; lacking theoretical grounded base from relying on the sorting/anti-sorting order for computation; lack of approximation error analysis; lack theoretical analysis for the approximation with finite number of directions instead of the continuous optimization problem; lack analysis with high-dimensional settings; weak experimental settings; lack the analysis of sensitivity to $L$; lack evaluation in high-dimensional settings beyond the current simple settings.

+ Reviewer pkqq: theoretical findings restricted with heavy assumptions; weak experimental settings; lacking large-scale experiments to illustrate its scalability

+ Reviewer d8H1: theoretical finding results restricted by strong assumptions; computational complexity?; theoretical results on invariance?

+ Reviewer LK3E: based on a false premise (i.e., closed-form expression of 1d-GW); empirical claims are not supported by theory (sorting/anti-sorting); novelty; limited insights from empirical finding results; weak experimental settings; misleading computational complexity

**Reviewer Scores:**

The authors clarify on theoretical findings on the definition and metric properties of the proposed MSGW; GAN experiments; computational complexity.

We think several concerns from the Reviewers still remains. The proposed approximation for the MSGW has not rigorously addressed in both theoretical and empirical aspects. It is questionable whether the theoretical finding results still hold for the discrete approximation. It is also better to provide rigorous proof for some important claims on invariance, especially its approximation.

The most critical one is that the submission is heavily based on a false premise (i.e., closed-form expression for 1d-GW)

---

### Decision · Program_Chairs · 2026-01-26

Reject